# Chemical and Pharmacological Profiling of *Wrightia coccinea* (Roxb. Ex Hornem.) Sims Focusing Antioxidant, Cytotoxic, Antidiarrheal, Hypoglycemic, and Analgesic Properties

**DOI:** 10.3390/molecules27134024

**Published:** 2022-06-22

**Authors:** Tabassum Jannat, Md. Jamal Hossain, Ahmed M. El-Shehawi, Md. Ruhul Kuddus, Mohammad A. Rashid, Sarah Albogami, Ibrahim Jafri, Mohamed El-Shazly, Mohammad Rashedul Haque

**Affiliations:** 1Phytochemical Research Laboratory, Department of Pharmaceutical Chemistry, Faculty of Pharmacy, University of Dhaka, Dhaka 1000, Bangladesh; tabassum_jannat@uap-bd.edu (T.J.); ruhulkuddus@du.ac.bd (M.R.K.); r.pchem@yahoo.com (M.A.R.); 2Department Pharmacy, University of Asia Pacific, Dhaka 1205, Bangladesh; 3Department of Pharmacy, State University of Bangladesh, 77 Satmasjid Road, Dhanmondi, Dhaka 1205, Bangladesh; 4Department of Biotechnology, College of Science, Taif University, P.O. Box 11099, Taif 21944, Saudi Arabia; elshehawi@hotmail.com (A.M.E.-S.); dr.sarah@tu.edu.sa (S.A.); i.jafri@tu.edu.sa (I.J.); 5Department of Pharmacognosy, Faculty of Pharmacy, Ain Shams University, Cairo 11566, Egypt; mohamed.elshazly@pharma.asu.edu.eg; 6Department of Pharmaceutical Biology, Faculty of Pharmacy and Biotechnology, German University in Cairo (GUC), Cairo 11835, Egypt

**Keywords:** *Wrightia coccinea*, antioxidant, cytotoxic, anti-diarrheal, hypoglycemic, analgesic, molecular docking

## Abstract

The aim of the study was to conduct phytochemical and pharmacological investigations of *Wrightia coccinea* (Roxb. ex Hornem.) Sims via several in vitro, in vivo, and in silico models. A total of four compounds were identified and isolated from the methanol extract of the bark and the methanol extract of the seed pulp of *W. coccinea* through successive chromatographic techniques and were characterized as 3*β*-acetyloxy-olean-12-en-28-ol (**1**), wrightiadione (**2**), 22*β*-hydroxylupeol (**3**), and *β*-sitosterol (**4**) by spectroscopic analysis. The aqueous fraction of the bark and chloroform fraction of the fruits provided the most potent antioxidant capacity (IC_50_ = 7.22 and 4.5 µg/mL, respectively) in DPPH free radical scavenging assay compared with the standard ascorbic acid (IC_50_ = 17.45 µg/mL). The methanol bark extract and the methanol fruit coat extract exerted anti-diarrheal activity by inhibiting 74.55 ± 0.67% and 77.78 ± 1.5% (mean ± SEM) of the diarrheal episode in mice, respectively, after four hours of loading the samples. In the hypoglycemic test, the methanol bark extract and the methanol fruit coat extract (400 mg/kg) produced a significant (*p* < 0.05) reduction in the blood glucose level in mice. Both doses of the plant extracts (200 mg/kg and 400 mg/kg) used in the study induced a significant (*p* < 0.05) increase in pain reaction time. The in vitro and in vivo findings were supported by the computational studies. The isolated compounds exhibited higher binding affinity compared with the standard drugs towards the active binding sites of glutathione reductase, epidermal growth factor receptor (EGFR), kappa opioid receptor, glucose transporter 3 (GLUT 3), Mu opioid receptor, and cyclooxygenase 2 (COX-2) proteins due to their potent antioxidant, cytotoxic, anti-diarrheal, hypoglycemic, and central and peripheral analgesic properties, respectively. The current findings concluded that *W. coccinea* might be a potential natural source for managing oxidative stress, diarrhea, hyperglycemia, and pain. Further studies are warranted for extensively phytochemical screening and establishing exact mechanisms of action.

## 1. Introduction

Medicinal plants containing bioactive molecules act as a vital resource of therapeutic agents for humankind [1,2,3]. Since ancient times, people have been exploring nature and searching for new therapeutic agents. This effort led to the discovery of many medicinal plants with healing action to treat various illnesses [4,5,6]. Nearly 80% of the population in the world depends on plant-based medicines as a significant source of healthcare [7,8,9]. The curative value of these plants lies in a wide variety of phytochemicals that exert diverse biological actions in the living system [10,11]. Many pharmaceuticals are derived from bioactive phytochemicals [12]. Antimalarial drugs such as quinine and artemisinin; the cardioactive drugs like digoxin and digitoxin; the narcotic analgesics like morphine; and anti-neoplastic agents like vincristine and vinblastine are obtained from medicinal plants. The primary phase of discovering potential leads from plant sources begins with phytochemical and pharmacological investigation of the plant samples to rationalize their traditional uses [13].

Apocynaceae, also known as the dogbane family, consists of flowering plants incorporating trees, shrubs, herbs, and vines [14]. *Wrightia coccinea* (Roxb. ex Hornem.) Sims, also known as *Scarlet wrightia* (Bengali name-Palan or Palam), is a small to medium-size deciduous tree which grows up to 8–10 m in height [15]. *W. coccinea* is a beautiful tree that grows as an ornamental plant beside parks and roadside areas. The plant is indigenous to Southeast Asia and is reported to be found in tropical Africa, China, the Indian Subcontinent, and Australia. In Bangladesh, the plant grows in mountainous regions of the Sylhet division. Appendix A presents different parts of the plant including the leaf, fruit, and seed. The flowers are attractive and scarlet in color. The species name “coccinea” comes from its crimson flowers [15]. The plant has blackish fruits with white spots. The plant is propagated by seeds that can produce cotton-like pulp. 

The species of *Wrightia* genus are very popular as an antidote to snakebite and are also used for dermatological purposes like psoriasis [16]. The famous species of this genus are *W. tinctoria*, *W. tomentosa*, and *W. rugosa. W. coccinea* is comparatively less investigated than other species but it was studied for its terpenoid constituents [16]. *W. tinctoria* seeds were popular as carminative, aphrodisiac, astringent, and tonic in India [17]. These species were also used for chest infections, especially asthma, colic, and diuresis [18]. Such diverse therapeutic properties of these species of the genus encouraged scientists to investigate other species. Some important chemical components like indole and triterpene were found in this plant including indigotin, indirubin, anthranilate, rutin, and tryptanthrin, which were also found in *W. tomentosa* [17]. The stem bark of *W. tinctoria* was found to contain *β*-amyrin, wrightiadione, wrightial, and lupeol [19,20]. Various studies demonstrated many promising biological activities of different parts of this plant including nociceptive, antidiabetic, anti-inflammatory, antiulcer, and anticancer activities [21]. The wide range of the genus therapeutic activity, as well as the diverse chemical constituents, directed our attention to a less investigated species, *W. coccinea*. Although the plant is traditionally used for several indications, there are no adequate data about the phytochemical content and pharmacological activity of *W. coccinea* to verify its traditional uses. Thus, we investigated the bark extract and seed pulp extract of *W. coccinea* to isolate and characterize the phytochemical constituents of this plant with subsequent assessment of the pharmacological properties of bark extract and fruit coat extract obtained from the plant. 

## 2. Materials and Methods

### 2.1. Plant Materials

Plant samples (bark, fruit coat, and seed pulp) of *W. coccinea* were collected from Bandarban Hill tracts in February 2018. They were taxonomically identified (Accession No. DACB-45689) by a scientific officer of Bangladesh National Herbarium, Mirpur 1, Dhaka-1216, Bangladesh (Appendix A).

### 2.2. Chemicals

All reagents and chemicals were of analytical grade. Lipophilic Sephadex LH-20 (Sigma Aldrich, Steinheim am Albuch, Germany), acetic acid (Merck, Darmstadt, Germany), Tween 80 (BDH Chemicals, UK), acetylsalicylic acid, loperamide, glibenclamide, morphine, and diclofenac sodium (Gonoshastho Pharmaceuticals Ltd., Dhaka, Bangladesh) were used for the study.

### 2.3. Extraction and Partitioning Process

The cleaned bark and fruit coat of the plant were sun-dried for 14 days while the seeds contained in the fruit generated cotton-like pulp. The dried samples were ground separately, and the pulverized samples of the bark, fruit coat, and seed pulp were extracted using Soxhlet apparatus [22]. The finely ground materials were packed in a ‘thimble’ made with filter paper and placed in a thimble chamber. The solvent (methanol) placed in the bottom chamber of the Soxhlet apparatus was then vaporized and allowed to be condensed and dripped on the thimble. The condensed solvent was collected in the bottom chamber containing the extracted compounds from the powdered material residing in the porous bag. Three different samples (bark, fruit coat, and seed pulp) were subjected to the process separately. The filtrate was condensed at reduced temperature and pressure using a rotary evaporator (Heidolph, Germany). The resulting % yield of crude methanol extract of the bark, fruit coat, and seed pulp was 17.83%, 15.38%, and 16.93%, respectively. Vacuum liquid chromatography (VLC) was employed for the fractionation of the plant extracts [23]. The column was packed with VLC grade silica and was washed with petroleum ether to ensure the compact packing. Approximately 15 g extracts of the bark and seed pulp were prepared by dissolving them into methanol and were mixed with silica, then dried. The dried sample of the bark extract (MEB) and seed pulp extract (MES) were applied separately to the top of the column, and elution was commenced starting with petroleum ether. The polarity of the eluting solvent was gradually increased by adding more polar solvents including ethyl acetate and methanol (Appendix A). The fractions 4–8 of VLC run were mixed together due to their identical characteristics and subjected to preparative TLC (stationary phase–silica gel PF_254_, mobile phase-ethyl acetate: petroleum ether = 5:95). The fractions of SEC were also mixed together due to their identical characteristics and subjected to preparative TLC. These compounds along with their sample ID are stated in Appendix A.

Both MEB and MEF were subjected to modified Kupchan partitioning into petroleum ether (PE), dichloromethane (DCM), chloroform (CF), and aqueous (AQ) soluble fractions [24]. All the plant samples were subjected to pharmacological evaluation.

### 2.4. Isolation of Chemical Compounds

The selected VLC fractions of the bark methanol extract were analyzed by gel permeation chromatography over lipophilic Sephadex LH-20 and PTLC over silica gel (F_254_). Following TLC screening of the chromatographic fractions and the subsequent PTLC analysis of the fractions, compounds **1**–**3** were isolated from the bark methanol extract (Figure 1). The chromatographic column separation of the seed pulp methanol extract and the subsequent PTLC of column fractions using ethyl acetate and toluene yielded compound **4** (Figure 1). PTLC was performed over silica gel 60_F254_ coated with glass plates, and 1% vanillin-sulfuric acid reagents were used to detect compounds. The NMR spectra of the isolated compounds were recorded in CDCl_3_ on Bruker 400 NMR machine.

### 2.5. In Vitro Antioxidant Activity: DPPH Assay

DPPH assay was used to estimate the antioxidant potential of a variety of plant samples using 2,2-diphenyl-1-picrylhydrazyl (DPPH) as a free radical [25,26,27]. It is noted that 2.0 mL solution of the plant sample at different concentrations (500.0 to 0.977 µg/mL) was mixed with 3.0 mL of DPPH solution in methanol, rendering the concentration of DPPH working solution of 20 µg/mL [28,29]. After 30 min incubation in the dark, the absorbance of each reaction mixture was recorded at 517 nm using a UV-visible spectrophotometer. The DPPH radical quenching capacity of *W. coccinea* was measured according to the following equation:$$\%\;\mathrm{Inhibition}=\frac{A_{\mathrm{blank}}-A_{\mathrm{sample}}}{A_{\mathrm{blank}}}\times 100\%$$
where A = absorbance for each group at 517 nm. The IC_50_ value (50% inhibition) for each tested sample was calculated from a plot of % inhibition of DPPH free radical vs. concentration (µg/mL) of the test materials.

### 2.6. In Vitro Cytotoxicity

Cytotoxic activity of the bark and fruit coat extracts was performed using brine shrimp lethality assay with vincristine sulfate (VS) as the positive standard [29,30]. A serial dilution of each tested sample (4 mg) in 99% DMSO (dimethyl sulfoxide) was prepared to obtain variable concentrations (400.0–0.781 µg/mL) of the tested solutions. Simulated seawater containing approximately ten live brine shrimp nauplii was added to each concentration of the tested solutions. After 24 h, the surviving nauplii were inspected by visual inspection with the help of a magnifying glass. For the individual concentration of the tested sample, the level of toxicity towards the shrimp was estimated by determining the LC_50_ value. LC_50_ value of the tested sample was determined from a plot of percentage of non-viable shrimps against the log concentration of plant extract using the standard curve of vincristine. 

### 2.7. Experimental Animals

Swiss-albino mice were used in the study. The mice of both sexes were collected from the Animal Resource Branch of the International Centre for Diarrheal Diseases and Research, Bangladesh (ICDDR,B). The mice weighed between 20–25 g and aged 4–5 weeks during the experiment. The standard polypropylene cages were utilized during housing the mice, and the temperature was controlled at 24 ± 2 °C with relative humidity 60–70% in the animal house, where a 12 h light-dark cycle was maintained. Because of the high susceptibility of the animals to environmental variation, all the acquired mice were kept in the laboratory environment for at least 3–4 days for acclimation. Standard guidelines for the care and use of laboratory animals were adopted while conducting the experiments. The Institutional Ethical Review Committee (SUB-IERC), State University of Bangladesh, has critically reviewed and approved the ethical issues. All detailed procedures and protocols of the study were approved, and an approval number (2018-08-13-SUB/A-ERC/007) was provided.

### 2.8. Preparation of the Loading Doses for In Vivo Study

For the in vivo model, the tested animals were given bark extract and fruit coat extract at doses of 200 and 400 mg/kg body weight. To prepare these loading doses, the accurately weighed samples were measured (24 mg on average for 200 mg/kg and 48 mg on average for 400 mg/kg) and were triturated in a unidirectional technique by mixing a small amount of 1% Tween 80 in normal saline. The volume of the suspension was made up to 3.0 mL. 

### 2.9. In Vivo Study Design

Mice were divided into the following groups, with four mice in each group. The negative and positive control group received 1% Tween 80 in normal saline (10 mL/kg) and standard drug, respectively. Group I and II were given 200 and 400 mg/kg of methanol extract of the bark, while Group III and IV received methanol extract of the fruit coat at 200 and 400 mg/kg, respectively.

### 2.10. Anti-Diarrheal Assay

The bark and fruit coat extracts were subjected to castor oil-induced diarrhea model in mice [31,32]. The positive control group received the standard loperamide and the negative control group received only 1% Tween 80 solutions. After half an hour of administering the sample and standard doses, 0.5 mL castor oil was administered to each animal orally with the help of a feeding needle. Each animal was separated into an individual cage. The floor was covered with a blotting paper that can hold clear stains of feces—they were observed for 4 h to monitor the anti-diarrheal effect of the tested samples. Data were collected every hour after castor oil administration. The percentage of inhibition of the defecation by the plant extracts was determined using the following formula
$$\%\;\mathrm{inhibition}\;\mathrm{of}\;\mathrm{defecation}=\frac{D_{\mathrm{control}}-D_{\mathrm{test}}}{D_{\mathrm{control}}}\times 100\%$$
where D = mean number of diarrheal episodes in each group.

### 2.11. Hypoglycemic Assay

The blood-glucose-lowering effect of the plant extract (200 and 400 mg/kg) was evaluated by oral glucose tolerance test according to the method described by Peungvicha et al. [33]. Six subgroups of animals were given sugar syrup orally with the help of a feeding needle. After 5 min, glibenclamide, water, 24 mg tested sample, and 48 mg tested sample were given to each animal of the positive control, negative control, Group-I, Group-II, Group-III, and Group-IV, respectively. Blood sugar was checked with diabetic strips every half an hour for 3 h. The % reduction in blood glucose level was calculated by the equation below:$$\%\;\mathrm{reduction}\;\mathrm{in}\;\mathrm{blood}\;\mathrm{glucose}=\frac{{\mathrm{BG}}_{\mathrm{control}}-{\mathrm{BG}}_{\mathrm{test}}}{{\mathrm{BG}}_{\mathrm{control}}}\times 100\%$$
where BG = average blood glucose level for each group.

### 2.12. Central Analgesic Activity

Tail immersion assay is a thermal process that was employed to assess the central analgesic activity of *W. coccinea extracts* [34]. The standard morphine (2 mg/kg, subcutaneous) solution was prepared by diluting the supplied morphine (15 mg/mL) with saline water [26]. The tested materials were orally administered utilizing a feeding needle to the mice. For the test, the mouse tail was immersed in hot water at 55 °C. The pain reaction time (PRT) or latency period for each mouse to flick its tail from the warm water was measured before (0 min) and at 0, 30, 60, and 90 min following the loading of the tested samples. 

### 2.13. Peripheral Analgesic Activity

Acetic acid-induced writhing method was employed to assess the peripheral analgesic activity of *W. coccinea extracts* [35]. The acetic acid-induced writhing method was applied to investigate the activity of the crude extract. Glacial acetic acid was used as a pain inducer to the animals of each subgroup. After an oral administration of the acid, aspirin and 24 and 48 mg sample was given to the positive control group, Group-I, Group-II, Group-III, and Group-IV, respectively, whereas the negative control group received Tween 80 solution alongside acetic acid. Writhing happens because of acetic acid. The number of writhes was measured for ten min after intraperitoneal injection of acetic acid. The percent inhibition of writhing was determined as follows
$$\%\;\mathrm{inhibition}\;\mathrm{of}\;\mathrm{writhing}=\frac{N_{\mathrm{Control}}-N_{\mathrm{Test}}}{N_{\mathrm{Control}}}\times 100\%$$
where N = mean number of abdominal writhing for each group.

### 2.14. Molecular Docking Study

Molecular docking analysis was performed to interpret the interactive profile of the four isolated compounds from *W. coccinea* with their target proteins. The widely used popular software packages, including PyRx, PyMoL 2.3, and BIOVA Discovery Studio version 4.5, were utilized during for the in silico study of the isolated compounds from *W. coccinea* according to the semiflexible procedures described in several studies [36,37,38,39,40,41].

#### 2.14.1. Target Protein Selection

Computational docking was conducted to reveal the potential bioactivities of the identified compounds including antioxidant, cytotoxicity, antidiarrheal, hypoglycemic, and analgesic properties. To run the molecular interaction and determine the radical scavenging capacity, cytotoxicity, antidiarrheal, hypoglycemic, and central and peripheral analgesic properties, glutathione reductase (PDB ID: 3GRS), epidermal growth factor receptor (PDB ID: 1XKK), kappa opioid receptor (PDB ID: 6VI4), glucose transporter 3 [GLUT 3] (PDB ID: 4ZWB), Mu-opioid receptor (PDB ID: 5C1M), and cyclooxygenase 2 (COX-2) (PDB ID: 1CX2) proteins were selected based on the biochemical mechanisms and the current evidence [36,42,43,44,45,46]. The targeted proteins’ three-dimensional (3D) crystal structures were retrieved from the RCSB protein data bank (https://www.rcsb.org; accessed on 1 January 2022) and were saved in PDB format. All collected biomolecules were opened in the PyMoL 2.3 software to delete water molecules and any unwanted residue from the proteins. Then the cleaned proteins were arranged by adding non-polar hydrogen atoms and were converted into the lowest energy state by applying Swiss PDB viewer for an energy minimization program. Finally, the cleaned and optimized proteins were saved in PDB format for further analysis. 

#### 2.14.2. Ligand Preparation

The structures of all isolated compounds (**1**–**4**) are presented in Figure 1. These four compounds 3β-acetyloxy-olean-12-en-28-ol (PubChem CID: 14010964), wrightiadione (PubChem CID: 10422105), 22β-hydroxylupeol (PubChem CID: 24786642), and β-sitosterol (PubChem CID: 222284), respectively, were searched and were found in PubChem database (https://pubchem.ncbi.nlm.nih.gov/; accessed on 1 January 2022). The 3D conformer of the ligands and the standard drugs, antioxidant acetyl salicylic acid (PubChem CID: 2244), vincristine (PubChem CID: 5978), loperamide (PubChem CID: 3955), glibenclamide (PubChem CID: 3488), morphine (PubChem CID: 5288826), and diclofenac (PubChem CID: 3033) were downloaded and were saved as SDF format. Then the ligands were loaded in the Discovery Studio version 4.5 serially and a ligand library was made with their PubChem CIDs in PDB format. All the phytoconstituents and the standard ligands were optimized via the Pm6 semi-empirical method for improving the accuracy of molecular interaction [37,47].

#### 2.14.3. Ligand Protein Interaction

Molecular docking was conducted to assume the potential binding profiles of the isolated phytoconstituents with their binding affinities towards the target macromolecules [47]. A widely used advanced software PyRxAutoDock Vina was applied for the drug-protein interaction, where a semiflexible modeling approach was adopted during the computer-aided docking process. The targeted protein was loaded and was selected as a macromolecule. The literature-based amino acids with their three-letter IDs were chosen for ascertaining the site-specific ligand-protein interaction. A total of eleven amino acids, VAL 102, LYS 127, ASN 129, VAL 130, GLN 131, LYS 143, SER 145, SER 147, GLY 148, ASP 183, and THR 185 were selected in glutathione reductase enzyme (PDB ID: 3GRS) for conducting active site-specific docking to predict the antioxidant effects of these isolated phytoconstituents [37]. To estimate the cytotoxicity, LEU 718, VAL 726, ALA 743, LYS 745, MET 766, LYS 775, ARG 776, LEU 777, LEU 788, THR 790, GLN 791, LEU 792, MET 793, GLY 796, CYS 797, LEU 799, ASP 800, ARG 803, LEU 844, THR 854, ASP 855, and PHE 856 were picked for the site-specific docking of the ligands with the epidermal growth factor receptor (PDB ID: 1XKK) [42]. LEU 103, LEU 107, SER 136, ILE 137, TRY 140, ILE 180, TRP 183, LEU 184, SER 187, ILE 191, LEU 192 ILE 194, and VAL 195 were traced during choosing the active sites of kappa opioid receptor (PDB ID: 6VI4) to dock with the ligands for projecting antidiarrheal potentiality [43]. The active sites of the GLUT3 (PDB ID: 4ZWB), Mu-opioid receptor (PDB ID: 5C1M), and cyclooxygenase 2 (COX-2) (PDB ID: 1CX2) proteins were selected based on the literature [44,45,46]. All the 3D conformers of the ligands (SDF format) were imported into the PyRx software and were run for the energy minimization of the ligands. All ligands were converted into pdbqt format in the PyRxAutoDock Vina software by utilizing Open Bable tool to equip the most suitable optimal hit. Then the grid box originated and the active binding sites of the proteins were kept within the center of the box, where the grid box mapping was as for center (X, Y, Z): (15.2267, 47.4301, 13.5499) and dimensions (angstrom) (X, Y, Z): (36.3425, 27.4717, 47.1464) during docking with 3GRS protein. The grid box mapping was fixed for 1KXX protein as center (X, Y, Z): (15.7032, 34.1446, 35.6152) and dimensions (angstrom) (X, Y, Z): (23.9494, 19.0224, 31.3996) where the grid box mapping was (X, Y, Z): (54.9442, −51.5736, −17.3795) and dimensions (angstrom) (X, Y, Z): (16.2278, 25.8672, 16.5909) for 6VI4 protein. All the receptors’ active binding sites were kept within the center space of the grid box, and grid box mapping values were recorded. The rest of the parameters during the docking process were set to the default settings. Docking was conducted under all stated conditions by employing AutoDock Vina (version 1.1.2). The results of the docking analysis were extrapolated, and the out files (pdbqt format) of the docked macromolecules and ligands were exported. The out files of the ligands and pdbqt file of the macromolecule were combined and were exported as PDB format via PyMol software for further visualization. The Discovery Studio Visualizer (version 4.5) was used for visualization and generating of 3D and 2D figures.

### 2.15. Statistical Analysis

Experimental data found from the in vitro and in vivo assays were reported as mean ± standard error of mean (SEM), whenever possible. Data were evaluated by Student’s *t*-test using GraphPad Software, USA. *p* values less than 0.05 were considered statistically significant. The molecular docking was conducted in triplicate, and the mean docking scores were presented in which the standard errors for all docking times were less than 1%. 

## 3. Results

### 3.1. Phytochemical Studies

We isolated four compounds including 3*β*-acetyloxy-olean-12-en-28-ol, wrightiadione, 22*β*-hydroxylupeol, and *β*-sitosterol (Figure 1). The structures of these compounds were determined and were confirmed comparing their NMR spectra with the reported data for similar compounds.

3*β*-Acetyloxy-olean-12-en-28-ol (Compound **1**): White solid crystal; ^1^H NMR (400 MHz, CDCl_3_): *δ* 5.30 (1H, br. s, H-12), 4.50 (1H, dd, *J* = 7.0, 10 Hz, H-3), 3.55 (2H, s, H_3_-28), 2.05 (3H, s, -OAc)*,* 1.16 (3H, s*,* H_3_-27)*,* 0.94 (6H, s, H_3_-29 and H_3_-30), 0.90 (3H, s, H_3_-25), 0.88 (3H, s, H_3_-26), 0.86 (3H, s, H_3_-23/H_3_-24) (Appendix A).

The ^1^H NMR spectrum (400 MHz, CDCl_3_) of compound **1** displayed a double doublet centered at *δ* 4.50, indicative of H-3 proton in a triterpene nucleus. The downfield shift of H-3 suggested that it was esterified. A singlet integration for three protons at *δ* 2.05 revealed the presence of an acetyl group at C-3. The ^1^H NMR spectrum of compound **1** displayed a one proton broad singlet at *δ* 5.30 for an olefinic proton at C-12. The ^1^H NMR spectral data of compound **1** demonstrated seven three proton singlets at *δ* 1.16, 0.94, 0.94, 0.90, 0.88, 0.86, and 0.86 for methyl groups. Thus, compound **1** was characterized as 3*β*-acetyloxy-olean-12-en-28-ol (Figure 1) [48].

Wrightiadione (Compound **2**): White amorphous powder; ^1^H NMR (400 MHz, CDCl_3_): *δ* 8.64 (1H, d*, J* = 8.0 Hz, H-8), 8.46 (1H, d, *J =* 8.0 Hz*,* H-5), 8.04 (1H, d, *J =* 8.0 Hz*,* H-3′), 7.93 (1H, d*, J* = 7.6 Hz*,* H-6′), 7.85 (1H, t, *J =* 7.6 Hz*,* H-5′), 7.79 (1H, t, *J =* 7.6 Hz, H-7), 7.68 (1H, t, *J* = 7.6 Hz, H-4′), 7.43 (1H, t, *J =* 7.6 Hz, H-6) (Appendix A).

The ^1^H NMR spectrum (400 MHz, CDCl_3_) of compound **2** gave resonances of four aromatic proton doublets at *δ* 8.64 (*J* = 8.0 Hz), 7.93 (*J* = 7.2 Hz), 8.04 (*J =* 8.0 Hz) and 8.46 (*J =* 8.0 Hz); four aromatic proton triplets at *δ* 7.79 (*J* = 1.2, 8.4 Hz), 7.43 (*J =* 7.2*,* 8.0 Hz), 7.85 (*J =* 7.2*,* 7.6 Hz), and 7.68 (*J* = 7.6, 8.0 Hz). The above signals revealed the presence of two aromatic rings with four adjacent protons in each ring, which indicated two isolated spin systems: H-5/H-6/H-7/H-8 and H-3′/H-4′/H-5′/H-6′. ^1^H NMR spectrum of compound **2** confirmed the aromatic rings without the hydroxy group. Compound **2** was characterized as wrightiadione [49].

22*β*-Hydroxylupeol (Compound **3**): White amorphous powder; ^1^H NMR (400 MHz, CDCl_3_): *δ* 4.69 (1H, m, H-29b), 4.56 (1H, m, H-29a), 3.64 (1H, t*, J* = 8.8 Hz, H-22), 3.20 (1H, dd*, J* = 5.6*,* 10.7 Hz, H-3), 2.36 (1H, m*,* H-19), 1.68 (3H, s, H_3_-30), 1.61 (1H, m, H-5), 1.07 (3H, s*,* H_3_-23), 1.03 (3H, s*,* H_3_-24), 0.94 (3H, s*,* H_3_-26), 0.87 and 0.83 (3H each, s, H_3_-28/H_3_-27), 0.79 (3H, s, H_3_-25) (Appendix A).

The ^1^H NMR data of compound **3** showed distinctive signals of lupine triterpene, seven methyl groups, one isopropenyl (*δ* 4.56, 4.69, and 1.68), and two signals of oxymethine hydrogens at *δ* 3.64 and 3.20. An extra oxymethine signal was also present, indicating that compound **3** is a hydroxylated lupeol derivative [50].

*β*-Sitosterol: White amorphous powder; ^1^H NMR (400 MHz, CDCl_3_): d 0.682 (3H,s, Me-18), 0.81 (3H, d, *J* = 7.0 Hz, Me-27), 0.83 (3H, d, *J* = 7.0 Hz, Me-26), 0.85 (3H,t, *J* = 7.0 Hz, Me-29), 0.92 (3H, d, *J* = 6.4 Hz, Me-21), 1.01 (3H, s, Me-19), 3.53 (1H, p, H-3a), 5.35 (1H,d br, *J* = 3.6 Hz, H6).

The spectroscopic data were identical to the published values [51]. The ^1^H NMR (400 MHz, CDCl_3_) of compound **4** (Figure 1) agreed with the published report of *β*-sitosterol and the spectrum was superimposable to the ^1^H NMR spectrum acquired for an actual sample formerly isolated in the laboratory [51]. The ^1^H NMR spectrum of **4** was provided in the Appendix A.

### 3.2. DPPH Free Radical Scavenging Activity

The tested samples showed quenching activities against DPPH radical in a concentration-dependent manner in the antioxidant assay. The IC_50_ values (50% inhibition) of the DPPH radical quenching capacity of the plant extracts are presented in Table 1. The smaller the IC_50_ values, and the higher the antioxidant potential of the tested sample. Among all tested samples, the chloroform fraction of the fruit coat extract and the aqueous fraction and dichloromethane fraction of the bark extract was the most potent DPPH radical scavenger, with IC_50_ values of 4.55, 7.22, and 10.91 µg/mL, respectively, in comparison with BHT (4.3 µg/mL) as the positive control. From the analysis of Table 1, we can conclude that the scavenging effect shown by different solvent fractions of the bark extracts on DPPH radicals were more potent than the fruit coat extracts of *W. coccinea*. 

### 3.3. Cytotoxic Activity

All the tested samples exhibited significant brine shrimp larvicidal activity in terms of LD_50_, which was comparable to the standard vincristine sulphate (LC_50_ = 0.451 µg/mL) (Table 1). The petroleum ether fraction of the fruit coat extract and the bark extract of *W. coccinea* displayed the maximum larvicidal activity with lethality concentrations (LC_50_) of 10.67 and 32.65 µg/mL, respectively. The second highest cytotoxic extracts were the dichloromethane and chloroform fraction of the bark extract and fruit coat extract of *W. coccinea*. 

### 3.4. Anti-Diarrheal Property

The tested fractions and standard loperamide demonstrated a significant (*p* < 0.05) and dose-dependent anti-diarrheal property in the animal study (Table 2). Castor oil-induced diarrhea persisted up to 4 h in the control group. This effect was diminished by loperamide (80.40 ± 0.61%) and by both bark extract and fruit coat extract of *W. coccinea* at dose 400 mg/kg with the maximum inhibitory effect of 74.55 ± 0.67% and 77.78 ± 1.5%, respectively.

### 3.5. Hypoglycemic Property

The hypoglycemic effects of the bark extract and fruit coat extract of *W. coccinea* in mice were summarized in Table 3. The tested samples of *W. coccinea* demonstrated significant (*p* < 0.05) and concentration-dependent glucose-lowering activity, which continued up to three hours following the loading dose. Both the bark extract and fruit coat extract exhibited promising hypoglycemic effects (74.7 ± 0.19% and 70.6 ± 0.30%, respectively) after 3 h while the standard glibenclamide produced 66.7 ± 0.61% reduction.

### 3.6. Central Analgesic Activity

The results of the central analgesic effect of the bark extract and fruit coat extract in the tail immersion method are shown in Table 4. All the tested samples showed a potent (*p* < 0.05) increase in the pain reaction time in comparison with the reference drug morphine (Table 4). Both the bark extract and fruit coat extract at a dose of 400 mg/kg body weight caused a significant increase in PRT, up to 8.50 ± 0.28 and 8.57 ± 0.19, respectively, compared with the standard morphine (12.06 ± 0.53) at 90 min after loading the plant sample. These extracts exhibited a potential analgesic effect in the experimental mice. 

### 3.7. Peripheral Analgesic Activity

The results of the peripheral analgesic effect of the bark extract and fruit coat extract (200 and 400 mg/kg body weight) in mice are summarized in Table 5. The plant extracts showed noticeable (*p* < 0.05) and concentration-dependent activity in reducing acetic acid-induced abdominal writhing in mice. The bark extract and fruit coat extract at the 400 mg/kg body weight dose exhibited 66.07 ± 0.88% and 54.39 ± 1.20% writhing inhibition in mice, respectively, compared with the standard diclofenac sodium (76.79 ± 0.33% inhibition).

### 3.8. In Silico Study

To understand the pharmacological activities of the extracts and different solvent fractions prepared from *W. coccinea,* molecular docking of the plant’s derived compounds against the corresponding molecular receptors was conducted by applying several suitable computer-based tools. All the docking scores collected from PyRx were tabulated in Table 6. The amino acid responsible for the interactions with the atom of the ligands, including the bond distance, bond type, and nature of the interaction, were tabulated in Appendix A. The lower the binding affinity (kcal/mol), the higher the binding strength. The extrapolated binding affinity having a null RMSD (root mean square deviation) value indicated the best docking prediction [37]. The enzyme/receptor inhibitory capacity of these isolated compounds was described as follows.

#### 3.8.1. Inhibition of Glutathione Reductase Enzyme: Antioxidant Activity

Glutathione reductase enzyme is associated with the regulation, modulation, and maintenance of redox homeostasis and oxidative stress [52]. All isolated compounds exerted good binding affinity (−8.4 to −9.3 kcal/mol; Table 6) towards the glutathione reductase enzyme compared with the standard antioxidant drug butylated hydroxytoluene (BHT) (−5.8 kcal/mol). The binding affinity order of the compounds were 22*β*-hydroxylupeol (compound **3**) > 3*β*-acetyloxy-olean-12-en-28-ol (compound **1**) > wrightiadione (compound **2**) = *β*-sitosterol (compound **4**) > BHT. The active binding sites of the glutathione reductase enzyme while interacting with the isolated compounds are summarized in Figure 2. A total of 13 hydrophobic interactions were noticed during the molecular docking of 22*β*-hydroxylupeol, where the number of alkyl and pi-alkyl interaction was eight and five, respectively. All the bonding sites including their corresponding distances were stated in Appendix A. The active binding sites for compound **1** were GLY 196, TYR 197, ILE 198, ALA 199, SER 225, PHE 226, ALA 288, ILE 289, GLY 290, ARG 291, ALA 336, LEU 337, LEU 338, THR 369, and THR 379 in A chain and for compound 2 were THR 57, VAL 61, GLY 62, CYS 63, LYS 66, SER 177, PHE 181, TYR 197, ILE 198, GLU 201, ARG 291, LEU 338, THR 339, and PRO 340. It is noted that the compound **1** formed H-bond (A:GLY290:HN-N:UNK1:O) with GLY 290 amino acid of the compound **1** and three hydrogen bonds (A:LYS66:HZ1-N:UNK1:O; A:LYS66:HZ3-N:UNK1:O; A:LYS66:HZ3-N:UNK1:O) were observed during interaction of compound **2** with LYS 66 amino acid of the protein (Appendix A). All the active binding sites of the compounds **3**, **4**, and standard drug BHT are summarized in Figure 2.

#### 3.8.2. Inhibition of EGFR: Cytotoxicity

All isolated compounds exhibited higher affinity (−8.1 to −9.4 kcal/mol; Table 6) towards the epidermal growth factor receptor (EGFR) compared with the standard drug vincristine (−6.3 kcal/mol). Among the four phytoconstituents, wrightiadione demonstrated the most cytotoxicity, where the active binding sites in A chain of the EGFR while interacting with the compound were LEU 718, VAL 726, ALA 743, LYS 745 THR 790 LEU 792, GLY 796, CYS 797, LEU 844, THR 854, and ASP 855. The interacting sites of the protein while docking with the isolated compounds are observed in Figure 3.

#### 3.8.3. Inhibition of Kappa Opioid Receptor: Antidiarrheal Activity

The interaction with the kappa opioid receptor (PDB ID: 6VI4) was studied. Compounds **1** (3*β*-acetyloxy-olean-12-en-28-ol) and **4** (*β*-sitosterol) showed the most antidiarrheal property via exhibiting the higher affinity (−7.7 kcal/mol) compared with the standard drug loperamide (−6.3 kcal/mol). Compounds **2** (wrightiadione) and **3** (22*β*-hydroxylupeol) exerted notable binding affinity (−7.1 kcal/mol and −7.0 kcal/mol, respectively). The order of the affinity of the compounds towards the receptor might be noted here as like 3*β*-acetyloxy-olean-12-en-28-ol = *β*-sitosterol > loperamide > wrightiadione (compound **2**) > 22*β*-hydroxylupeol. Most of the binding interaction sites are hydrophobic in nature. However, compound **1** exerted two hydrogen bonds (B:SER136:HG -N:UNK1:O; B:TRP183:CD1-N:UNK1:O) at 2.39948 and 3.38888 Å with the SER 136 and TRP 183 amino acids of the protein (Appendix A). The active binding amino acids of the receptor were PHE 99, LEU 103, LEU 107, ILE 133, SER 136, ILE 137, TYR 140, TRP 183, SER 187, ASN 179, and ILE 180 during interacting with compound 1 and ILE 96, PHE 99, ASN 100, LEU 103, LEU 107, SER 136, TYR 140, LYS 176, ASN 179, ILE 180, TRP 183, SER 187, and ILE 194 when docked with compound **4**. Similarly, all the interacting amino acids with their three letters IDs are summarized in Figure 4.

#### 3.8.4. Inhibition of GLUT 3: Hypoglycemic Activity

The molecular docking scores of the four compounds (**1–4**) were found as −8.6, −9.5, −8.6, and −9.4 kcal/mol, respectively, in the interaction with the GLUT 3 receptor. The active binding sites in A chain of the receptor were THR 15, THR 18, ILE 19, OHE 22, LEU 169, GLY 161, VAL 164, PHE 190, LEU 193, and PRO 194 during docking with compound **1**. Besides, PHE 24, THR 28, ASN 32, GLN 159, ILE 162, ILE 166, GLN 280, GLN 281, ILE 285, ASN 286, PHE 289, TYR 290, PHE 377, TRP 386, and ASN 413 in A chain were found as the active binding sites while interacting with compound **2** (wrightiadione) which showed the most hypoglycemic property among the isolated phytoconstituents. All the active binding sites of the protein during docking with the reported compounds are summarized in Figure 5.

#### 3.8.5. Inhibition of Mu-Opioid Receptor and COX-2 Proteins: Analgesic Activity

Molecular docking of the isolated compounds with the Mu-opioid receptor and COX-2 protein was performed to reveal the molecular mechanism of analgesic activity of the *W. coccinea* extracts. All compounds exerted higher binding affinity towards COX 2 enzyme (−8.5 to −9.6 kcal/mol) compared with the standard analgesic agent, diclofenac (−7.0 kcal/mol). The order of the docking scores of the compounds against COX-2 proteins were *β*-sitosterol > wrightiadione > 3*β*-acetyloxy-olean-12-en-28-ol > 22*β*-hydroxylupeol > diclofenac (Table 6). Wrightiadione and *β*-sitosterol showed interaction with Mu-opioid receptor (−9.1 kcal/mol and −9.7 kcal/mol, respectively) than the standard drug morphine (−8.0 kcal/mol). The active binding sites of Mu-opioid receptor during docking with *β*-sitosterol (compound **4**) were HIS 54, SER 55, GLN 124, ASN 127, TRP 133, ASP 147, TYR 148, MET 151, LEU 232, LYS 233, VAL 236, PHE 237, TRP 293, ILE 296, HIS 297, VAL 300, TRP 318, ILE 322, and TYR 326 in chain A of the protein. The active sites were LEU 15, ASN 34, CYS 36, ASN 39, CYS 41, GLN 42, ASN 43, ARG 44, GLY 45, GLU 46, CYS 47, GLY 135, PRO 153, ALA 156, GLN 461 GLU 465, LYS 468, and ARG 469, when *β*-sitosterol interacted with the COX 2 enzyme. The active binding sites of the Mu-opioid receptor and COX 2 enzyme are found in Figure 6 and Figure 7 during interaction with the isolated phytoconstituents from the plant extract. 

## 4. Discussion

Medicinal plants are gifted sources of bioactive phytochemicals, which possess various pharmacological properties. The isolation and characterization of phytochemicals is an established way of ascertaining the bioactive profile of a medicinal plant. In phytochemical analysis, various chromatographic techniques such as CC, TLC, HPLC, GC, and PTLC can be used to isolate and purify bioactive compounds from plant extracts. In our study, we extracted *W. coccinea* with methanol and followed gel permeation chromatographic technique, TLC screening, and subsequent PTLC analysis. Several pure compounds (Figure 1) were obtained from the bark and seed pulp extracts of *W. coccinea*.

The current study reported the isolation of four compounds from *W. coccinea* including 3*β*-acetyloxy-olean-12-en-28-ol, wrightiadione, 22*β*-hydroxylupeol, and *β*-sitosterol [45,46,47,48]. According to the best of our knowledge, this is the first investigation on *W. coccinea* to report these bioactive phytoconstituents. Nevertheless, we found several plants of the genus *Wrightia,* where several terpenes (for example, lupeol) and sterols (for example, β-sitosterol) were reported from *W. tinctoria* [53,54]. Additionally, a recent study reported wrightiadione from *W. hanleyi* [55].

Several pharmacological activities of the plant extracts were evaluated to search for new therapeutic potentials of the plant and validate its folk medicinal use. The antioxidant property of *W. coccinea* was investigated in the search for safe and effective antioxidant candidates from natural resources. The antioxidant capacity of this plant species was assayed in terms of estimation the total phenol content and the DPPH radical quenching activity. *W. coccinea* bark and fruit coat extracts exhibited promising antioxidant potential compared with the reference antioxidant BHT and ascorbic acid. A moderate level of the phenolic content was also observed in different solvent fractions of the bark and fruit coat extracts. The plant-derived phenolic compounds serve as antioxidants through many possible pathways [56,57,58,59,60]. The hydroxyl group of the polyphenols scavenge reactive free radicals and thereby protect the biological system from free radical-induced oxidative stress [61]. All isolated compounds revealed higher affinities towards the glutathione reductase enzyme than the standard compound BHT. Among these molecules, 22*β*-hydroxylupeol showed the maximum free radical scavenging property in the molecular docking study. The compound exerted direct DPPH scavenging and membrane permeability, revealing strong antioxidant properties [62]. All the hydrophobic interactions of the compound with the glutathione reductase enzyme through alkyl and pi-alkyl interactions might be responsible for such actions. Sudhahar et al. [63] demonstrated that lupeol decreased lipid peroxidation (LPO) and showed enzymatic and nonenzymatic antioxidant capacity in animal models by reducing oxidative cellular damage. This might be due to the higher hydrophobic interactions with the glutathione reductase enzyme (Appendix A). *β*-Sitosterol is also responsible for minimizing the detrimental effect of free radicals like peroxynitrite and inhibiting the LPO and NO generation [64,65].

As toxicity is a major concern for a crude drug, brine shrimp (*Artemia salina*) lethality bioassay, a cost-effective and reliable technique for preliminary screening of cytotoxicity, was conducted to predict the toxicity of the plant extracts [66]. According to the study performed by Logarto Parra et al. [67], there is a good correlation (r = 0.85; *p* < 0.05) between the 50% lethal concentration (LC_50_) found in brine shrimp assay and 50% lethal dose (LD_50_) obtained from an animal study, suggesting the brine shrimp test can be used as an alternative model. Meyer et al. [68] reported that the cutoff point for bioactive phytochemicals is LC_50_ value less than 1000 μg/mL. The current study revealed that the measured LC_50_ values in the brine shrimp lethality bioassay were below 1000 μg/mL. No crude extract and fraction derived from the plant might be considered severely toxic or lethal, endorsed by the findings of the previous studies (LC_50_ > 10 μg/mL) [66,69]. All the isolated compounds from the plant were molecularly docked with the EGFR protein, which has a significant role in cellular signal transduction and cell survival [70]. The overproduction of EGFR is strongly associated with several types of cancer progression like ovarian cancer, breast cancer, and colon cancer [71]. Thus, EGFR inhibition might be an exciting target for developing selective anticancer molecules [42,72]. All the isolated phytoconstituents exerted more potent cytotoxic potential and safety than the standard vincristine. All the plant-derived compounds exhibited 1 to 3 conventional H-bond formation (except *β*-sitosterol) and 11 to 20 hydrophobic interactions and showed better fitting to the active sites of EGFR (Appendix A). Notably, compound 2 (wrightiadione) exhibited the highest potent EGFR protein inhibition (binding affinity = −9.4 kcal/mol), exerted as a promising candidate for anticancer drug development. The abundance of hydrophobic interactions of the compound with the protein through pi-alkyl and pi-sigma might be responsible for the actions. Recently, scientists have reported that wrightidione and its derivatives showed promising anticancer potentillas through inhibiting tropomyosin-related kinases (Trks) [73,74].

Since ancient times, medicinal plants have been used to manage several gastro-intestinal (GI) related disorders, including diarrhea. However, the safety and efficacy profiles of the major part of these plants have not been investigated. Therefore, this study was designed to appraise the safety and efficiency of *W. coccinea* as an anti-diarrheal potential option in the management of diarrheal illness in mice. Ricinoleic acid, the chief constituent of castor oil, is reported to irritate the gut wall, which can bring out peristaltic movement and finally induces diarrhea [75]. The both crude extracts prepared from the bark and fruit coat of *W. coccinea* showed notable antidiarrheal properties in a dose-dependent manner. In the case of fruit coat methanolic extract, the lower dose extract showed no significant antidiarrheal effect because the lower dose had not enough capacity to prolong the onset of diarrhea [75]. This study demonstrated that the antidiarrheal activity of the bark and fruit coat extracts of *W. coccinea* might be attributed to the bioactive phytochemicals responsible for the antidiarrheal effect [76]. All compounds isolated from the plant extracts showed comparable binding affinity towards kappa opioid receptor (−7.0 to −7.7 kcal/mol), supporting the antidiarrheal effect of the plant extracts. *β*-Sitosterol, an isolated secondary metabolite from *W. coccinea*, might play a significant role in reducing the peristalsis in the GI system and thus prevents GI motility [75,77].

Diabetes mellitus, a metabolic disorder, is now prevailing as a universal public health problem. Therefore, the demand for safe and effective antidiabetic molecules from natural sources intensifies day by day. Due to the low toxicity, affordable price, and availability of plants, herbal medicines can be widely used to manage hyperglycemia. Several reports were published explaining the blood-glucose-lowering effect of the plant extracts. Therefore, the present work is concerned with the preliminary assessment of the hypoglycemic effect of *W. coccinea* in animals. Our study revealed that the soluble fractions of the bark and fruit coat extracts of *W. coccinea* exhibited significant lowering of blood glucose after glucose-induced hyperglycemia in mice. Therefore, the significant hypoglycemic effect of *W. coccinea* produced in this analysis could be attributed to the presence of phytochemicals [78]. All the isolated compounds exhibited a notable binding affinity towards GLUT 3 in the computational docking study. This is a novel investigation of the plant extract and thus there are no previous data regarding the hypoglycemic property of the plant. However, the petroleum ether extract (400 mg/kg) of *W. tinctoria* L. showed almost 75% glucose-lowering capacity after 14 days in alloxan-induced diabetic rats [79]. The secondary metabolites, *β*-sitosterol and lupeol or its derivatives, isolated from the *W. coccinea,* showed a significant antidiabetic activity in several studies [80,81]. Many studies reported that oxidative stress is directly associated with the etiology and pathogenesis of several disorders like cancer and diabetes mellitus [82,83]. As all the compounds showed higher binding affinity (−8.4 to −9.3 kcal/mol) towards the glutathione reductase enzyme than the standard BHT (−5.8 kcal/mol) and comparable inhibition property of glucose transporter 3 (GLUT 3) (−8.6 to 9.4 kcal/mol), these four compounds (compounds 1 to 4) could be potentially considered the responsible molecules isolated from *W. coccinea* having glucose-lowering property [64,84]*.*

Pain-relieving agents are being searched from natural sources as alternatives to synthetic drugs because they have fewer adverse effects [85]. In our study, the analgesic activity of methanol extract of the bark and methanol extracts of the fruit coat was evaluated by tail immersion method and writhing method in mice. In our experimental condition, the plant samples significantly reduced both heat-induced and acetic acid-induced pain sensation in mice. Our study suggested that the bark and fruit coat extracts of *W. coccinea* may contain phytochemicals that can reduce pain sensation by inhibiting prostaglandin synthesis [86,87]. Furthermore, the compounds found in the plant extract exhibited more potent affinities towards the COX 2 protein than the standard drug diclofenac in the computational study supporting the in vivo findings. Furthermore, the *β*-sitosterol and wrightiadione exerted higher binding affinity than the standard morphine through conventional hydrogen bond and hydrophobic interactions. Nirmal et al. [88] reported that the isolated compound *β*-sitosterol exhibited analgesic activity in hot plate and acetic acid-induced assays by inhibiting central opioid receptors or facilitating the discharge of endogenous opioid peptides and blocking the production of prostaglandins and bradykinins.

## 5. Conclusions

The phytochemical analysis of bark extract and seed pulp extract of *W. coccinea* afforded four compounds characterized as 3β-acetyloxy-olean-12-en-28-ol, wrightiadione, 22β-hydroxylupeol, and β-sitosterol by spectroscopic techniques. The isolated plant metabolites may be functional for medicinal purposes. In biological screening, the chloroform soluble fraction of fruit coat extract of *W. coccinea* showed maximum DPPH radical quenching capability as an antioxidant agent with an IC_50_ value of 4.55 μg/mL compared to the standard ascorbic acid (IC_50_ = 17.45 μg/mL). The in vivo assay results demonstrated that the bark extract and fruit coat extract of the plant species possess outstanding anti-diarrheal and analgesic properties. The bark extract and fruit coat extract exerted promising glucose-lowering capacity (74.7 ± 0.19% and 70.6 ± 0.30% of blood glucose level reduction, respectively) after three hours, while the standard drug glibenclamide produced 66.7 ± 0.61% of blood glucose level reduction. Furthermore, the in-silico investigations supported the in vitro and in vivo results as most of the isolated compounds exhibited promising binding affinities towards the corresponding receptors. Thus, the present pharmacological study unveiled the effectiveness of *W. coccinea* as a hopeful source of potential bioactive molecules that may be considered for novel drug discovery and therapeutic progress. Nonetheless, further studies are imperative to discover the bioactive compounds from *W. coccinea* extracts accountable for these bioactivities. 

## Figures and Tables

**Figure 1 molecules-27-04024-f001:**
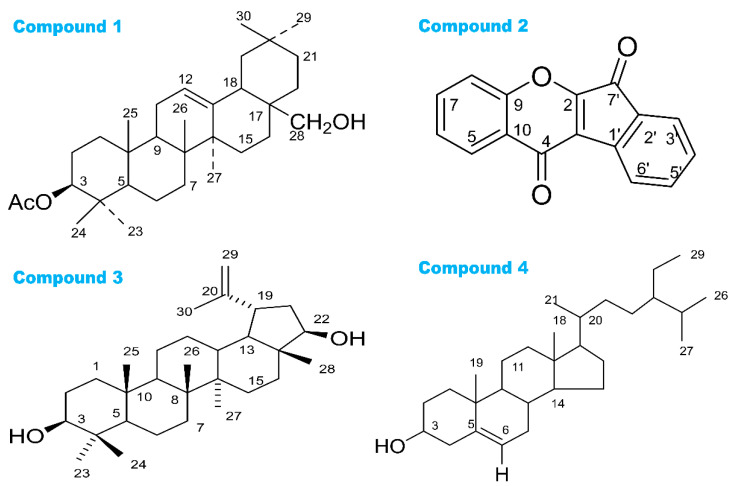
The structures of isolated four compounds (compound **1**: 3*β*-Acetyloxy-olean-12-en-28-ol; compound **2**: Wrightiadione; compound **3**: 22*β*-hydroxylupeol; and compound **4**: *β*-sitosterol) from *W. coccinea*.

**Figure 2 molecules-27-04024-f002:**
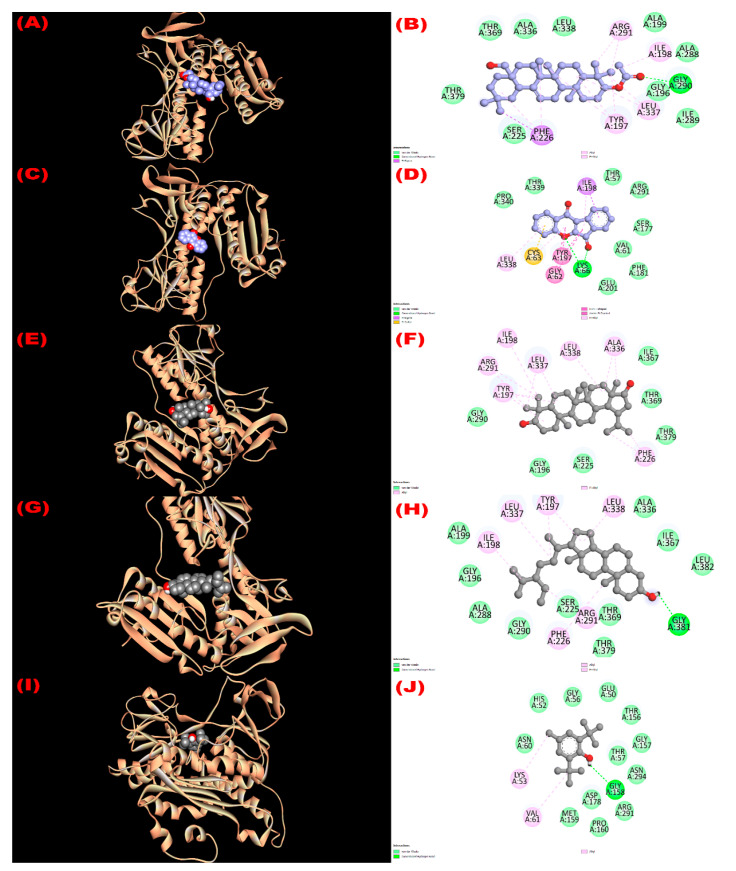
The 3D and 2D molecular interactions of the isolated compounds with the glutathione reductase enzyme (PDB ID: 3GRS) revealing the antioxidant potentiality. ((**A**,**B**), (**C**,**D**), (**E**,**F**), (**G**,**H**), (**I**,**J**) sketched the pictorial view of molecular docking of compounds (**1** to **4**) 3*β*-acetyloxy-olean-12-en-28-ol, Wrightiadione, 22*β*-hydroxylupeol, *β*-sitosterol, and the standard BHT, respectively).

**Figure 3 molecules-27-04024-f003:**
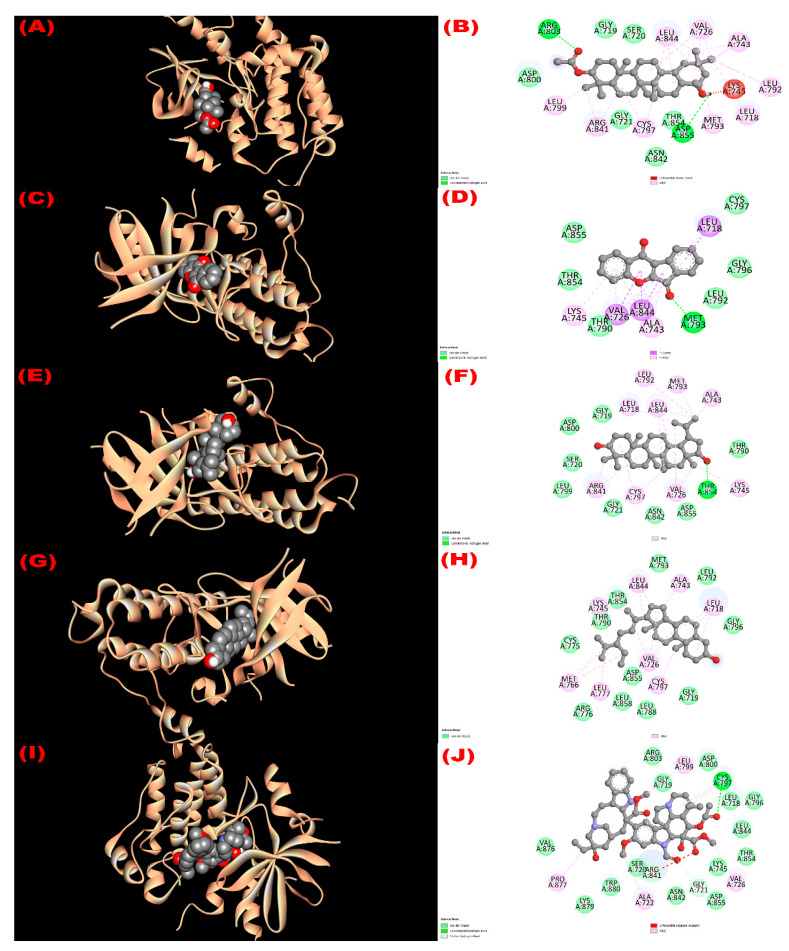
The 3D and 2D view of molecular interactions of the isolated compounds with the epidermal growth factor receptor (PDB ID: 1XKK) revealing the cytotoxicity. ((**A**,**B**), (**C**,**D**), (**E**,**F**), (**G**,**H**), (**I**,**J**) sketched the pictorial view of molecular interactions of compounds (**1** to **4**) 3*β*-acetyloxy-olean-12-en-28-ol, Wrightiadione, 22*β*-hydroxylupeol, *β*-sitosterol, and the standard vincristine, respectively).

**Figure 4 molecules-27-04024-f004:**
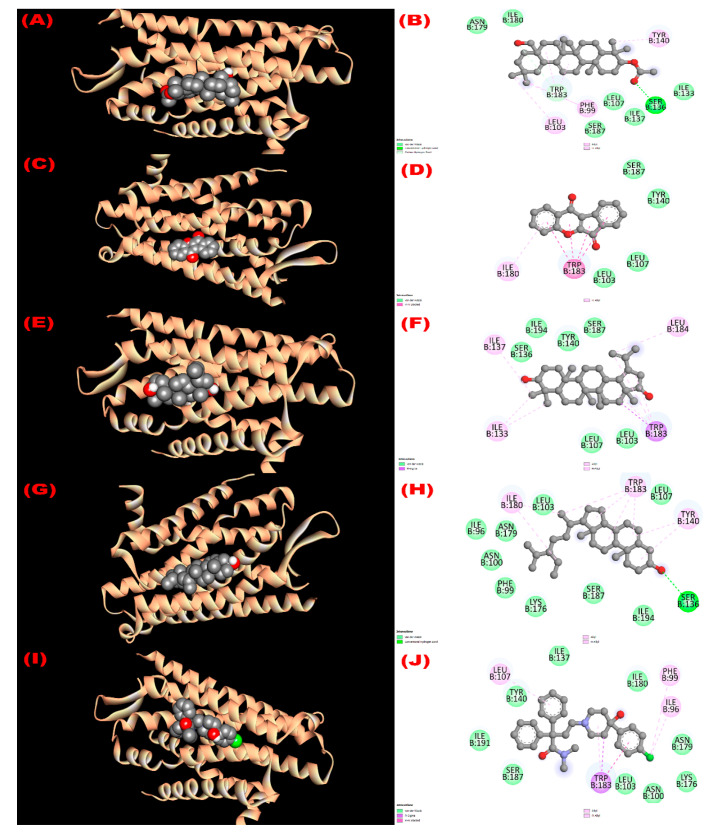
The 3D and 2D view of molecular interactions of the isolated compounds with the kappa opioid receptor (PDB ID: 6VI4) revealing the antidiarrheal activity. ((**A**,**B**), (**C**,**D**), (**E**,**F**), (**G**,**H**), (**I**,**J**) sketched the pictorial view of molecular interactions of compounds (**1** to **4**) 3*β*-acetyloxy-olean-12-en-28-ol, Wrightiadione, 22*β*-hydroxylupeol, *β*-sitosterol, and the standard loperamide, respectively).

**Figure 5 molecules-27-04024-f005:**
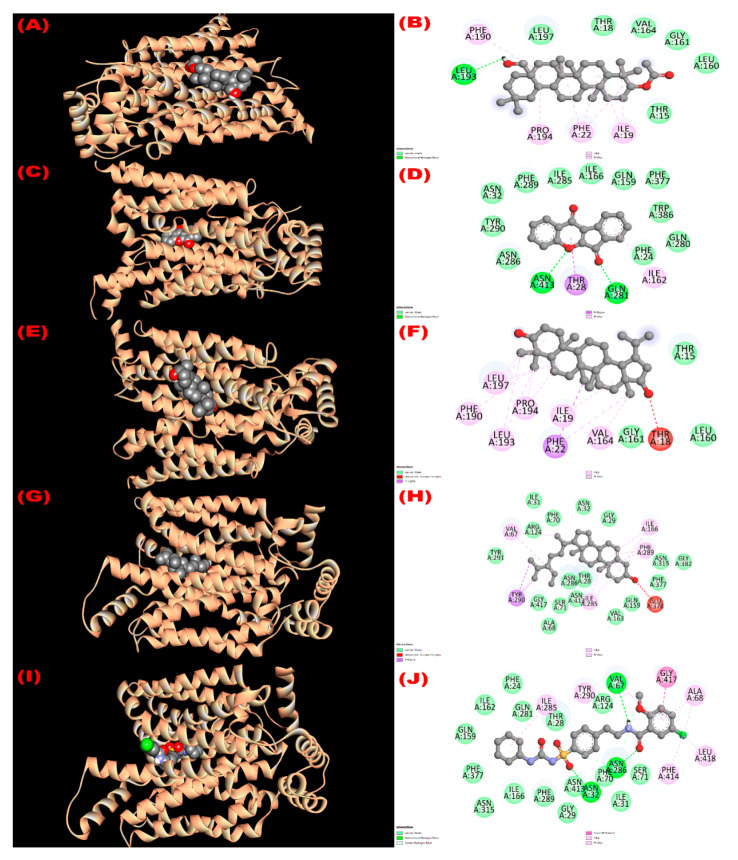
The 3D and 2D view of molecular interactions of the isolated compounds with the glucose transporter 3 (GLUT 3) (PDB ID: 4ZWB) revealing the hypoglycemic activity. ((**A**,**B**), (**C**,**D**), (**E**,**F**), (**G**,**H**), (**I**,**J**) sketched the pictorial view of molecular interactions of compounds (**1–4**) 3*β*-acetyloxy-olean-12-en-28-ol, wrightiadione, 22*β*-hydroxylupeol, *β*-sitosterol, and the standard glibenclamide, respectively).

**Figure 6 molecules-27-04024-f006:**
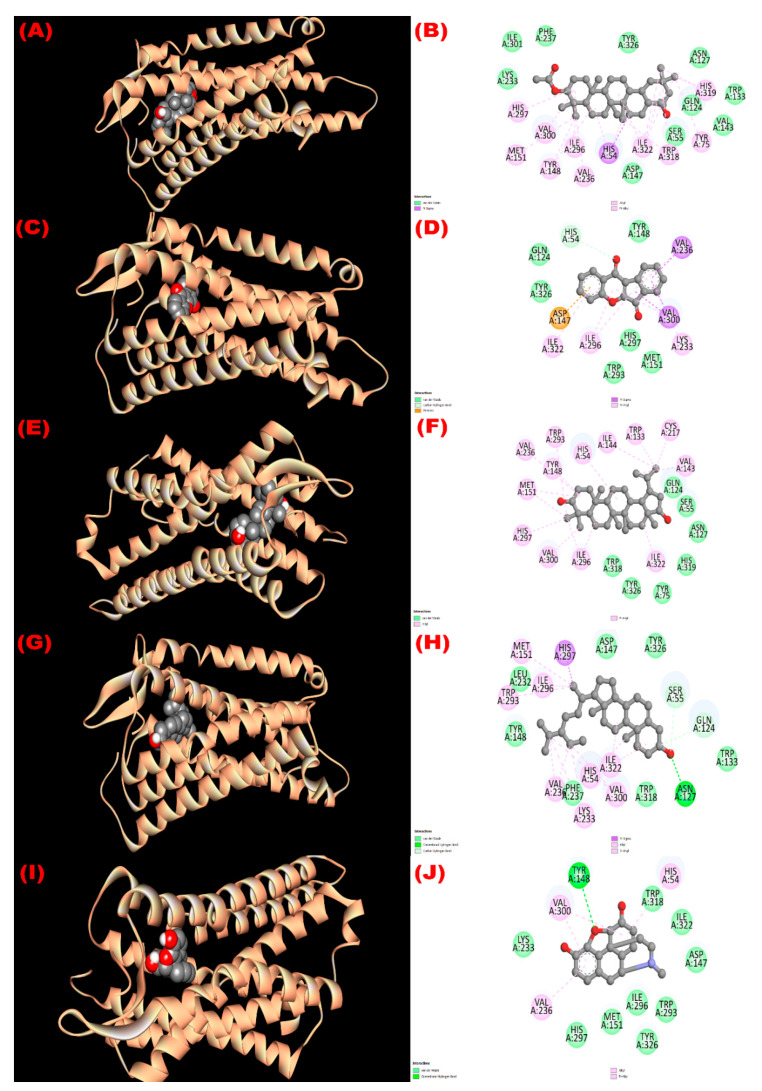
The 3D and 2D view of molecular interactions of the isolated compounds with the Mu-opioid receptor (PDB ID: 5C1M) revealing the central analgesic activity. ((**A**,**B**), (**C**,**D**), (**E**,**F**), (**G**,**H**), (**I**,**J**) sketched the pictorial view of molecular interactions of compounds (**1–4**) 3*β*-acetyloxy-olean-12-en-28-ol, wrightiadione, 22*β*-hydroxylupeol, *β*-sitosterol, and the standard morphine, respectively).

**Figure 7 molecules-27-04024-f007:**
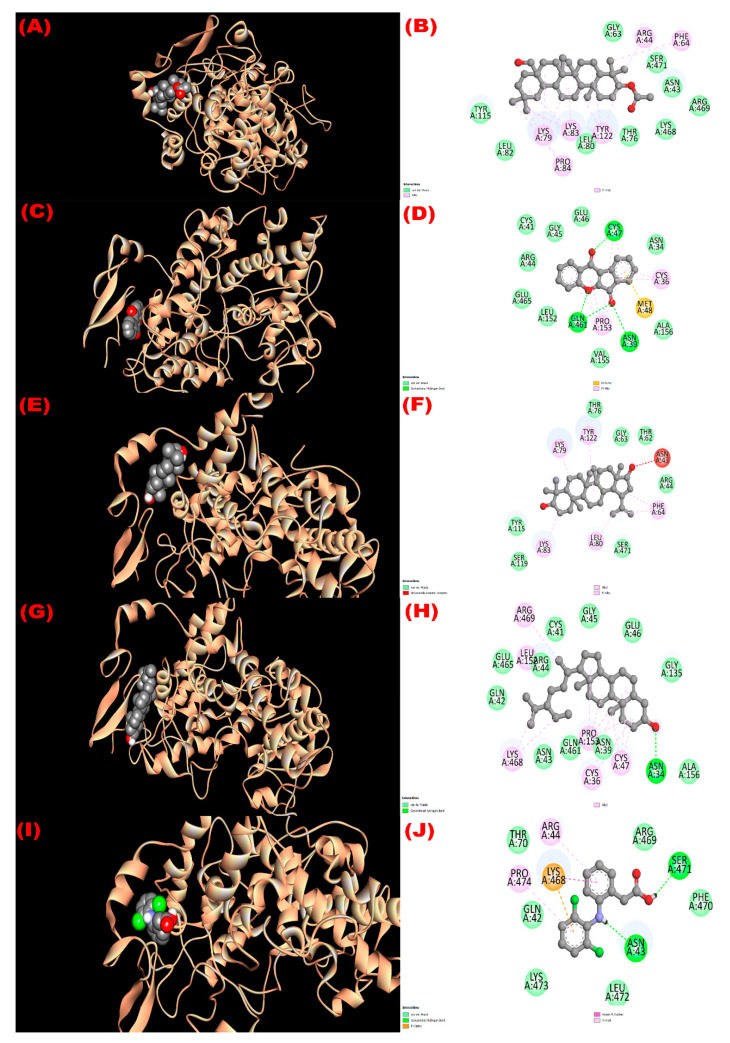
The 3D and 2D view of molecular interactions of the isolated compounds with the cyclooxygenase 2 (COX-2) (PDB ID: 1CX2) revealing the peripheral analgesic activity. ((**A**,**B**), (**C**,**D**), (**E**,**F**), (**G**,**H**), (**I**,**J**) sketched the pictorial view of molecular interactions of compounds (**1–4**) 3*β*-acetyloxy-olean-12-en-28-ol, wrightiadione, 22*β*-hydroxylupeol, *β*-sitosterol, and the standard diclofenac, respectively).

**Table 1 molecules-27-04024-t001:** Antioxidant potentials and cytotoxic effects with IC_50_ (obtained from DPPH assay) and LC_50_ (obtained from brine shrimp lethality assay) values of MEB, MEF, and different solvent fractions of *W. coccinea*.

Sample Code	IC_50_ (µg/mL) from DPPH Assay		LC_50_ (µg/mL) from Brine Shrimp Lethality Assay	
	Bark	Fruit Coat	Bark	Fruit Coat
ME	90.86	31.6	161.50	160.91
PE	79.95	118.9	32.65	10.67
DCM	10.91	337.05	41.05	41.08
CF	26.88	4.55	40.39	41.51
AQ	7.22	35.25	51.09	70.47
BHT	4.3	4.3		
AA	17.45	17.45		
VS			0.451	0.451

Here, ME = methanol extract of the bark (MEB) or fruit coat (MEF) of *W. coccinea*; PE, DCM, CF, AQ = petroleum ether, dichloromethane, chloroform, and aqueous soluble fraction of either the methanol extract of bark (MEB) or the methanol extract of fruit coat (MEF) of *W. coccinea*, respectively. BHT = butylated hydroxytoluene; AA = ascorbic acid; VS = vincristine sulphate.

**Table 2 molecules-27-04024-t002:** Anti-diarrheal effects in terms of % reduction of diarrheal feces following administrating the methanolic extract of bark (MEB) and methanolic extract of fruit (MEF) of *W. coccinea* in castor oil-induced diarrheal mice model.

Group	Treatment	% Reduction in Diarrheal Feces			
		Time after Loading the Plant Sample/Drug			
		1 h	2 h	3 h	4 h
Positive control	Loperamide 50 mg/kg	100 ± 0.0 **	88.89 ± 0.33 **	82.95 ± 0.20 *	80.40 ± 0.61 *
I	MEB 200 mg/kg	71.43 ± 0.33 *	66.67 ± 0.58	64.19 ± 0.67	48.18 ± 1.0
II	MEB 400 mg/kg	85.71 ± 0.33 **	87.78 ± 0.67 **	80.2 ± 0.33 *	74.55 ± 0.67
III	MEF 200 mg/kg	69.35 ± 0.33	58.67 ± 0.58	50.0 ± 0.67	53.33 ± 1.0
IV	MEF 400 mg/kg	88.69 ± 0.33 **	80.40 ± 0.67 *	79.21 ± 0.78	77.78 ± 1.5

Data are mean ± SEM for *n* = 4, ** *p* < 0.01 and * *p* < 0.05 vs. control.

**Table 3 molecules-27-04024-t003:** Hypoglycemic effect in terms of % reduction in blood glucose level of methanolic extract of bark (MEB) and methanolic extract of fruit coat (MEF) of *W. coccinea* on glucose-induced hyperglycemic mice model.

Group	Treatment	% Reduction in Blood Glucose Level of Mice		
		Time after Loading the Plant Sample/Drug		
		1 h	2 h	3 h
Positive control	Glibenclamide 10 mg/kg	29.1 ± 0.79	62.1 ± 0.21 *	66.7 ± 0.61 **
I	MEB 200 mg/kg	30.0 ± 0.36	53.1 ± 0.63	60.1 ± 0.87 *
II	MEB 400 mg/kg	43.8 ± 1.14	46.2 ± 1.20	74.7 ± 0.19 **
III	MEF 200 mg/kg	30.0 ± 0.36	53.1 ± 0.63	60.1 ± 0.78
IV	MEF 400 mg/kg	26.6 ± 1.47	43.3 ± 0.56	70.6 ± 0.30 *

Data are mean ± SEM for *n* = 4, ** *p* < 0.01 and * *p* < 0.05 vs. control.

**Table 4 molecules-27-04024-t004:** Central analgesic effect of methanolic extract of bark (MEB) and methanolic extract of fruit coat (MEF) of *W. coccinea* in mice by tail immersion method.

Group	Treatment	Average Time of Tail Immersion of Mice		
		Time (in Sec) after Loading the Plant Sample/Drug		
		30 min	60 min	90 min
Negative control	Tween 80 solution	2.50 ± 0.05 **	2.41 ± 0.11 **	2.18 ± 0.13 **
Positive control	Morphine 2 mg/kg	5.42 ± 0.21 **	10.20 ± 0.66 **	12.06 ± 0.53 **
I	MEB 200 mg/kg	3.67 ± 0.16 *	5.69 ± 0.08 **	6.92 ± 0.39 **
II	MEB 400 mg/kg	4.16 ± 0.06 **	6.58 ± 0.40 **	8.50 ± 0.28 **
III	MEF 200 mg/kg	3.99 ± 0.14 **	5.94 ± 0.23 **	7.24 ± 0.46 **
IV	MEF 400 mg/kg	4.28 ± 0.24 *	6.20 ± 0.10 **	8.57 ± 0.19 **

Data are mean ± SEM for *n* = 4, ** *p* < 0.01 and * *p* < 0.05 vs. control.

**Table 5 molecules-27-04024-t005:** Peripheral analgesic effect of methanolic extract of bark (MEB) and methanolic extract of fruit coat (MEF) of *W. coccinea* in acetic acid-induced abdominal writhing mice.

Group	Treatment	% Inhibition of Writhing
Negative control	Tween 80 solution	--
Positive control	DS 50 mg/kg	76.79 ± 0.33 **
I	MEB 200 mg/kg	42.86 ± 0.88 *
II	MEB 400 mg/kg	66.07 ± 0.88 **
III	MEF 200 mg/kg	45.61 ± 1.20 **
IV	MEF 400 mg/kg	54.39 ± 1.20 **

DS = Diclofenac sodium. Data are mean ± SEM for *n* = 4, ** *p* < 0.01 and * *p* < 0.05 vs. control.

**Table 6 molecules-27-04024-t006:** Molecular docking scores or binding affinity (kcal/mol) retrieved from in silico interactions of the isolated compounds from *W. coccinea* (Roxb. ex Hornem.) Sims and the standard drugs during the interaction with glutathione reductase (PDB ID: 3GRS), epidermal growth factor receptor (PDB ID: 1XKK), kappa opioid receptor (PDB ID: 6VI4), glucose transporter 3 (GLUT 3) (PDB ID: 4ZWB), Mu-opioid receptor (PDB ID: 5C1M), and cyclooxygenase 2 (COX-2) (PDB ID: 1CX2) proteins for assessing the antioxidant, cytotoxicity, antidiarrheal, hypoglycemic, and central and peripheral analgesic activities, respectively.

Com. No.	Name of Compounds/Drugs	PubChem ID	Binding Affinity towards Corresponding Receptors/Macromolecules (kcal/mol)					
			3GRS (Antioxidant)	1XKK (Cytotoxicity)	6VI4 (Antidiarrheal)	4ZWB (Hypoglycemic)	5C1M (Central Analgesic)	1CX2 (Peripheral Analgesic)
**1**	3β-acetyloxy-olean-12-en-28-ol	14010964	**−9.0**	**−8.1**	**−7.7**	−8.6	−6.2	**−8.6**
**2**	Wrightiadione	10422105	**−8.4**	**−9.4**	−7.1	−9.5	**−9.1**	**−9.4**
**3**	22*β*-Hydroxylupeol	24786642	**−9.3**	**−9.2**	−7.0	−8.6	−6.8	**−8.5**
**4**	*β*-Sitosterol	222284	**−8.4**	**−9.2**	**−7.7**	−9.4	**−9.7**	**−9.7**
Standard drugs	Butylated hydroxy toluene (BHT)	31404	−5.8					
	Vincristine	5978		−6.3				
	Loperamide	3955			−7.3			
	Glibenclamide	3488				−10.2		
	Morphine	5288826					−8.0	
	Diclofenac	3033						−7.0

## Data Availability

All the raw data of this research can be obtained from the corresponding authors upon reasonable request.

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
