# Peer review of "Chemical and Pharmacological Profiling of Wrightia coccinea (Roxb. Ex Hornem.) Sims Focusing Antioxidant, Cytotoxic, Antidiarrheal, Hypoglycemic, and Analgesic Properties"

_molecules, 2022, doi:10.3390/molecules27134024_

Round 1

Reviewer 1 Report

The manuscript entitled “Phytochemical Isolation of Wrightia coccinea (roxb. Ex hornem.) Sims and Evaluation of its Potential Pharmacological Properties Via In Vitro, In Vivo and In Silico Approaches” presents a significant research work. But manuscript requires revision on basis of following points

  • Generally, the manuscript needs extensive English improvements.
  • The title of this manuscript is not attractive, Please modify this
  • Section Material and method: subsection extraction and partioning process line # 105 Please mention which solvent you have used in bottom chamber for extraction?
  • Line # 103. Please mention which three samples you are writing about. This is not clear?
  • Line 112. There is need to mention how much amount in grams of plant parts were used for extraction process which produced the yield 83.4 g, 79.1 g, and 50.8 g, ?? Also calculate % yield please.
  • There is no need to cite ref 25, because self citations are ethically wrong and also for methods, authors must cite the original reference for method section.
  • Line 128. It states that authors have isolated 1-3 compounds, but results and discussion section stating 1-4. Please correct this statement as per results
  • Why authors did not perform the biological activities of 1-4 isolated compounds?? If authors have carried out the bioassays of crude extracts, this is not peculiar, If possible then authors must perform the bioassays of pure isolated molecules
  • Please mention quantitative results of your studies in conclusion section.
  • Please provide us letter of ethical committee approval regarding animal study.
  • For NMR section. Authors have provided proton NMR, Authors must provide Carbon NMR of isolated compounds as well in discussion section and also provide spectra in supplementary information.
  • For in silico studies, discussion is weak, please follow this paper, and use this as reference in your discussion part to make strong effect; doi: 10.1016/j.bioorg.2019.103216.
  • In case of antioxidant studies and cytotoxic studies, to broader the scope of studies please include following literature:

https://link.springer.com/article/10.1186/s13065-018-0495-1

https://www.mdpi.com/1422-0067/13/5/6440

https://biolres.biomedcentral.com/articles/10.1186/0717-6287-47-12

doi: 10.1186/1752-153X-8-12.

  • In my opinion, the introduction lacks a clear structure. The overview from previous relevant research (previous related references) should be incorporated in introduction section. To arouse a broad interest from readership in this field, some recent literature should be included i.e.,

doi: 10.3390/molecules171214275

https://doi.org/10.1016/j.arabjc.2013.04.024

https://doi.org/10.3390/ijms22147463

Author Response

Reviewer 1

The manuscript entitled “Phytochemical Isolation of Wrightia coccinea (roxb. Ex hornem.) Sims and Evaluation of its Potential Pharmacological Properties Via In Vitro, In Vivo and In Silico Approaches” presents a significant research work. But manuscript requires revision on basis of following points

Authors’ responses: I would like to thank you very much to reviewer 1 for reviewing the manuscript and providing your expert comments and opinions on the manuscript. All the comments were welcomed and carefully studied. We have improved and corrected the raised points accordingly. Thank you again for your kind efforts and time for reviewing the manuscript.  

  • Generally, the manuscript needs extensive English improvements.

Authors’ responses: Thank you very much for the comments. We have revised and done the English improvements of the manuscript.

  • The title of this manuscript is not attractive, Please modify this

Authors’ responses: Thank you very much for comments and suggestions. We have modified the title as follows:

Isolation of Phytoconstituents of Wrightia coccinea (roxb. Ex hornem.) Sims and Evaluation of its Antioxidant, Cytotoxicity, Antidiarrheal, Hypoglycemic, and Analgesic Properties Via In Vitro, In Vivo and In Silico Approaches (Lines: 1-5)

  • Section Material and method: subsection extraction and partioning process line # 105 Please mention which solvent you have used in bottom chamber for extraction?

Authors’ responses: Thank you for the query. We have revised as follows: 

The solvent (methanol) placed in the bottom chamber of the Soxhlet apparatus was then vaporized and allowed to be condensed and dripped on the thimble. (Lines: 109-111)

  • Line # 103. Please mention which three samples you are writing about. This is not clear?

Authors’ responses: Thank you for the comment. We have revised as follows: 

Three different samples (bark, fruit coat, and seed pulp) were subjected to the process separately. (Lines: 113-114)

  • Line 112. There is need to mention how much amount in grams of plant parts were used for extraction process which produced the yield 83.4 g, 79.1 g, and 50.8 g, ?? Also calculate % yield please.
  • Authors’ responses: Thank you for the comment and suggestion. We have deleted the line regarding the amount of the crude extracts, whereas we calculated % yield as follows:

The resulting % yield of crude methanol extract of the bark, fruit coat, and seed pulp were 17.83%, 15.38%, and 16.93%, respectively. (Lines: 115-116)

  • There is no need to cite ref 25, because self citations are ethically wrong and also for methods, authors must cite the original reference for method section.

Authors’ responses: Thank you for the comment & suggestions. We should have been more careful during citations. We have replaced the 25-number citation with the following references. Also, we have confirmed the original references in method section in most of the cases.   

  • Line 128. It states that authors have isolated 1-3 compounds, but results and discussion section stating 1-4. Please correct this statement as per results.

Authors’ responses: Thank you very much for the nice suggestions. Thank you very much for the comments. Kindly be informed that compounds 1-3 were isolated from the bark methanol extract and compound 4 was isolated form seed pulp extract. We have revised to clear understanding the line as follows:

Following TLC screening of the chromatographic fractions and the subsequent PTLC analysis of the fractions, compounds 1-3 were isolated from the bark methanol extract (Figure 1). The chromatographic column separation of the seed pulp methanol extract and the subsequent PTLC of column fractions using ethyl acetate and toluene yielded compound 4 (Figure 1). (Lines: 131-134)

  • Why authors did not perform the biological activities of 1-4 isolated compounds?? If authors have carried out the bioassays of crude extracts, this is not peculiar, If possible then authors must perform the bioassays of pure isolated molecules.

Authors’ responses: Thank you very much for your comments and suggestions. We have investigated the biological assays of the bark and fruit methanolic extracts of W. coccinea focusing antioxidant, cytotoxicity, antidiarrheal, hypoglycemic, and analgesic properties. It is true that we did not test the bioassays of these isolated compounds in this study via in vitro and in vivo experiments due to the less amount of the compounds during the experiments. However, we have extensively done the in-silico analysis of these compounds to clearly understand the binding affinity of these isolated compounds towards the corresponding receptors. The in-silico studies might be considered the quick analysis to interpret the responsible bioactivity of the extracts.

  • Please mention quantitative results of your studies in conclusion section.

Authors’ responses: Thank you very much for the nice suggestions. We have revised and added quantitative results of our studies in conclusion part as follows:  

The isolated plant metabolites may be functional for medicinal purposes. In biological screening, the chloroform soluble fraction of fruit coat extract of W. coccinea showed maximum DPPH radical quenching capability as an antioxidant agent with an IC50 value of 4.55 μg/mL compared to the standard ascorbic acid (IC50 = 17.45 μg/mL). (Lines: 703-707)

The bark extract and fruit coat extract exerted promising glucose-lowering capacity (74.7 ± 0.19% and 70.6 ± 0.30% of blood glucose level reduction, respectively) after three hours, while the standard drug glibenclamide produced 66.7 ± 0.61% of blood glucose level reduction. (Lines: 709-712)

  • Please provide us letter of ethical committee approval regarding animal study.

Thank you very much for your comment. Please find the picture of the approval letter regarding animal study.

  • For NMR section. Authors have provided proton NMR, Authors must provide Carbon NMR of isolated compounds as well in discussion section and also provide spectra in supplementary information.

Authors’ responses: I would like to thank you very much for the comments and suggestions. We have isolated these compounds and confirmed the purification through TLC. Then we have taken the proton NMR spectrum and searched literature. We have found the same compounds in the literature and confirmed the structure by the reported published data. That’s why, we did not take the carbon NMR of the compounds. 

  • For in silico studies, discussion is weak, please follow this paper, and use this as reference in your discussion part to make strong effect; doi: 10.1016/j.bioorg.2019.103216.

Authors’ responses: Thank you very much for the comments and sharing the source. We have carefully studied the shared article and found several indications for the improvement of the discussion part. Consequently, we have cited it in the discussion part. We have also carefully improved the discussion part as follows in respect to your comments:

……………….All the hydrophobic interactions of the compound with the glutathione reductase enzyme through alkyl and pi-alkyl interactions might be responsible for such actions. (Lines: 634-636)

……………… Notably, compound 2 (wrightiadione) exhibited the highest potent EGFR protein inhibition (binding affinity = -9.4 kcal/mol), exerted as a promising candidate for anticancer drug development. The abundance of hydrophobic interactions of the compound with the protein through pi-alkyl and pi-sigma might be responsible for the actions. Recently, scientists have reported that wrightidione and its derivatives showed promising anticancer potentillas through inhibiting tropomyosin-related kinases (Trks) [73,74] (Lines: 661-666)

……….. Many studies reported that oxidative stress is directly associated with the etiology and pathogenesis of several disorders like cancer and diabetes mellitus [82,83]. As all the compounds showed higher binding affinity (-8.4 to -9.3 kcal/mol) towards the glutathione reductase enzyme than the standard BHT (-5.8 kcal/mol) and comparable inhibition property of glucose transporter 3 (GLUT 3) (-8.6 to 9.4 kcal/mol), these four compounds (compounds 1 to 4) could be potentially considered the responsible molecules isolated from W. coccinea having glucose-lowering property [84,85]. (Lines: 692-699)

  • In case of antioxidant studies and cytotoxic studies, to broader the scope of studies please include following literature:

https://link.springer.com/article/10.1186/s13065-018-0495-1

https://www.mdpi.com/1422-0067/13/5/6440

https://biolres.biomedcentral.com/articles/10.1186/0717-6287-47-12

doi: 10.1186/1752-153X-8-12.

Authors’ responses: Thank you very much for the comments and sharing the links, which were very helpful. We have cited and improved the discussion part accordingly.

  • In my opinion, the introduction lacks a clear structure. The overview from previous relevant research (previous related references) should be incorporated in introduction section. To arouse a broad interest from readership in this field, some recent literature should be included i.e.,

doi: 10.3390/molecules171214275

https://doi.org/10.1016/j.arabjc.2013.04.024

https://doi.org/10.3390/ijms22147463

Authors’ responses: Thank you very much for the comments and suggested articles’ links, which were very helpful. We have cited these articles in part introduction part accordingly.

Reviewer 2 Report

Review reports molecules-1680918-R1:

This manuscript's title is "Phytochemical Isolation of Wrightia coccinea (roxb. Ex hornem.) Sims and Evaluation of its Potential Pharmacological Properties Via In Vitro, In Vivo and In Silico Approaches" The authors intended to investigate The isolated compounds of Wrightia coccinea exhibited higher binding affinity towards the active binding sites of glutathione reductase, epidermal growth factor receptor (EGFR), kappa opioid receptor, glucose transporter 3 (GLUT 3), Mu opioid receptor, and cyclooxygenase 2 (COX-2) proteins due to their potent antioxidant, cytotoxic, anti-diarrheal, hypoglycemic, central and peripheral analgesic properties, respectively. However, some notes below that the authors must explain well.

Comments from the reviewer:

  1. Suggest a conclusion section at the end of the Abstract.
  2. There were no results about seed pulp extract (MES). Suggesting cancel all the described sections related to MES fraction.
  3. β-acetyloxy-olean-12-en-28-ol (1), wrightiadione (2), 22β-hydroxylupeol (3) and β-sitosterol (4) were isolated from Wrightia coccinea. Suggesting the authors supplement the yield rate of each compound, isolated protocol or data (PTLC, VLC, or HPLC), and explain why did not use them or β-sitosterol (easy to obtain) as a positive control in vitro or in vivo study?
  4. In table 2, the authors have separated the methanolic extract (ME) into Petroleum ether (PE), dichloromethane (DCM), chloroform (CF), and aqueous (AQ) fraction. Which fraction contains those four compounds? Why did not use the bioactive fraction but used the crude methanolic extract in vivo study?
  5. Those four compounds were not phenolic compounds and could be not the chemical antioxidative compounds, why did the authors measure the total phenolic content assay and DPPH assay? It is the wrong direction. Otherwise, those pharmacological active compounds were from phenolic compounds, and those activities were not from those four compounds? Suggest the authors must explain well.
  6. In table 1, It is unbelievable, that IC50 was higher than LC50. What is the possibility that the authors can get those results? Suggesting the authors must explain them well.
  7. According to the pharmacotherapies concept, the therapeutic index LC50/IC50 is better lower than 10. The LC50 of MEB and MEF were very lower. It’s meaning that MEB and MEF seem to have a very high toxicity. Do they worthy to introduce to people? Suggesting the authors must explain them well.
  8. MEB and MEF have different IC50. Why did the authors use 200 mg/kg and 400 mg/kg for the animal test? How those dosages were established and based on? Suggest the authors must explain them well.
  9. All the legends of the Tables did not describe well. Suggest must supplement experimental protocol, animal number, and statistical analysis.

Author Response

Reviewer 2.

Comments and Suggestions for Authors

Review reports molecules-1680918-R1:

This manuscript's title is "Phytochemical Isolation of Wrightia coccinea (roxb. Ex hornem.) Sims and Evaluation of its Potential Pharmacological Properties Via In Vitro, In Vivo and In Silico Approaches" The authors intended to investigate The isolated compounds of Wrightia coccinea exhibited higher binding affinity towards the active binding sites of glutathione reductase, epidermal growth factor receptor (EGFR), kappa opioid receptor, glucose transporter 3 (GLUT 3), Mu opioid receptor, and cyclooxygenase 2 (COX-2) proteins due to their potent antioxidant, cytotoxic, anti-diarrheal, hypoglycemic, central and peripheral analgesic properties, respectively. However, some notes below that the authors must explain well.

Authors’ responses: Thank you very much to reviewer 2 for reviewing the manuscript and providing your comments and opinions on the manuscript. All the comments were carefully studied. We have improved and corrected your raised points accordingly. Thank you again for your kind efforts and time for reviewing the manuscript. 

Comments from the reviewer:

  1. Suggest a conclusion section at the end of the Abstract.

Authors’ responses: Thank you very much for your kind suggestion. We have added the following lines in the abstract part.   

The current findings concluded that W. coccinea might be a potential natural source for managing oxidative stress, diarrhea, hyperglycemia, and pain. Further studies are warranted for extensively phytochemical screening and establishing exact mechanisms of action. (Lines: 42-45)     

  1. There were no results about seed pulp extract (MES). Suggesting cancel all the described sections related to MES fraction.

Authors’ responses: Thank you very much for your comments. We have isolated three compounds (1 to 3) from methanol extract of bark and another compound was isolated from methanol extract of seed pulp (compound 4).

  1. β-acetyloxy-olean-12-en-28-ol (1), wrightiadione (2), 22β-hydroxylupeol (3) and β-sitosterol (4) were isolated from Wrightia coccinea. Suggesting the authors supplement the yield rate of each compound, isolated protocol or data (PTLC, VLC, or HPLC), and explain why did not use them or β-sitosterol (easy to obtain) as a positive control in vitro or in vivo study?

Authors’ responses: Thank you very much for your nice observations and comments. We have stated as follows and added two tables in supplementary file (Table S1 and Table S2).

The polarity of the eluting solvent was gradually increased by adding more polar solvents including ethyl acetate and methanol (Table S1). The fractions 4-8 of VLC run were mixed together due to their identical characteristics and subjected to preparative TLC (stationary phase – Silica gel PF254, Mobile Phase - Ethyl acetate: Petroleum ether = 5:95). The fractions of SEC were also mixed together due to their identical characteristics and subjected to preparative TLC. These compounds along with their sample ID are stated in Table S2. (Lines 129-135)

Table S1. Different solvent fraction used for VLC analysis of crude methanol extract of bark of W. coccinea

Beaker Number

Solvent System

Volume Collected

1

100 % Hexane

150 mL

3

2.5 % Ethyl Acetate in Hexane

150 mL

4

5 % Ethyl Acetate in Hexane

150 mL

5

7.5 % Ethyl Acetate in Hexane

150 mL

6

10 % Ethyl Acetate in Hexane

150 mL

7

12.5 % Ethyl Acetate in Hexane

150 mL

8

15 % Ethyl Acetate in Hexane

150 mL

9

17.5% Ethyl Acetate in Hexane

150 mL

10

20 % Ethyl Acetate in Hexane

150 mL

11

22.5% Ethyl Acetate in Hexane

150 mL

12

25 % Ethyl Acetate in Hexane

150 mL

13

27.5% Ethyl Acetate in Hexane

150 mL

14

30 % Ethyl Acetate in Hexane

150 mL

15

35 % Ethyl Acetate in Hexane

150 mL

16

40 % Ethyl Acetate in Hexane

150 mL

17

45 % Ethyl Acetate in Hexane

150 mL

18

50 % Ethyl Acetate in Hexane

150 mL

19

60 % Ethyl Acetate in Hexane

150 mL

20

70 % Ethyl Acetate in Hexane

150 mL

21

80 % Ethyl Acetate in Hexane

150 mL

22

100 % Ethyl Acetate

100 mL

23

5 % Methanol in Ethyl Acetate

100 mL

24

10 % Methanol in Ethyl Acetate

100 mL

25

20 % Methanol in Ethyl Acetate

100 mL

26

30 % Methanol in Ethyl Acetate

100 mL

Table S2. Preparative Thin Layer Chromatography (PTLC) of selected fractions from Size Exclusion Chromatography (SEC) of different VLC fractions of W. coccinea.

Fraction

Solvent System

UV

Spray Color

Sample ID

Short

Long

4-8 (beaker direct from VLC)

Ethyl acetate: Petroleum ether= 5:95)

Yes

No

Brown

Vial-A

Yes

No

Violet

WC-1

No

Yes

Ash

WC-2

9-16

Ethyl acetate: Hexane= 5:95)

No

No

Violet

WC-3

No

Yes

Yellow

WC-4

Yes

No

Yellow

WC-5

Yes

No

Pink

WC-6

17-22

Chloroform: Hexane= 80:20)

No

No

Violet

WC-11

23-28

Chloroform: Hexane= 80:20)

No

No

Purple

WC-7

No

Yes

Violet

WC-8

Yes

Yes

-

WC-9

Yes

Yes

-

Vial-37

29-33

Chloroform: Hexane= 80:20)

Yes

No

Dark violet

Vial-38

No

Yes

-

WC-10

Again, we did not use beta sitosterol as positive control rather we have used beta sitosterol  as in investigatory compound from W. coccinea.

  1. In table 2, the authors have separated the methanolic extract (ME) into Petroleum ether (PE), dichloromethane (DCM), chloroform (CF), and aqueous (AQ) fraction. Which fraction contains those four compounds? Why did not use the bioactive fraction but used the crude methanolic extract in vivo study?

Authors’ responses: Thank you very much for your comments. We have clearly mentioned in method section that we have isolated three compounds 1 to 3 from the methanolic bark extract and compound 4 from the methanolic seed pulp extract of the plant. Therefore, in the animal study (in vivo investigation), we have used the crude extract. 

  1. Those four compounds were not phenolic compounds and could be not the chemical antioxidative compounds, why did the authors measure the total phenolic content assay and DPPH assay? It is the wrong direction. Otherwise, those pharmacological active compounds were from phenolic compounds, and those activities were not from those four compounds? Suggest the authors must explain well.

Authors’ responses: Thank you very much for your comments. We have agreed with your comments that all these four compounds are not chemically phenolic compounds. We have investigated the total phenolic content (TPC) from the fractions derived from the fruit and bark methanolic extracts, and we have found promising results in the TPC and DPPH assay. Though we have isolated four compounds, and they are not phenolic. However, the aqueous fraction of the bark and chloroform fraction of the fruits provided the maximum amount of phenol content (220.62 and 190.83 mg of GAE/g of the dry extract, respectively). They also revealed the most potent antioxidant capacity (IC50 = 7.22 and 4.5 µg/mL, respectively) in DPPH free radical scavenging assay compared with the standard ascorbic acid (IC50 = 17.45 µg/mL). These results indicated that the fractious contain more antioxidant compounds and phenolic compounds that may have created a new indication for future prospective research. Besides, the in-silico studies revealed that the four compounds have more binding affinity towards glutathione reductase enzyme compared to the standard antioxidant drug BHT.  

  1. In table 1, It is unbelievable, that IC50 was higher than LC50. What is the possibility that the authors can get those results? Suggesting the authors must explain them well.

Authors’ responses: Thank you very much for your observations and comments. The IC50 values of the fractions were measured from the DPPH assay and LC50 values were measured from the brine shrimp lethality bioassay. For your kind perusal, we have stated here the modified legend of the table 1 as follows:

Table 1: Total phenol content, antioxidant potentials, and cytotoxic effects with IC50 (obtained from DPPH assay) and LC50 (obtained from brine shrimp lethality assay) values of MEB, MEF and different solvent fractions of W. coccinea. (Lines: 398-399)

  1. According to the pharmacotherapies concept, the therapeutic index LC50/IC50 is better lower than 10. The LC50 of MEB and MEF were very lower. It’s meaning that MEB and MEF seem to have a very high toxicity. Do they worthy to introduce to people? Suggesting the authors must explain them well.

Authors’ responses: Thank you very much for your nice observations and comments. We have addressed and discussed the issue in the discussion part of the manuscript as follows:

As toxicity is a major concern for a crude drug, brine shrimp (Artemia salina) lethality bioassay, a cost-effective and reliable technique for preliminary screening of cytotoxicity, was conducted to predict the toxicity of the plant extracts [1]. Meyer et al. [2] reported that the cutoff point for bioactive phytochemicals is LC50 value less than 1000 μg/mL. The current study revealed that the measured LC50 values in the brine shrimp lethality bioassay were below 1000 μg/mL. No crude extract and fraction derived from the plant might be considered severely toxic or lethal, endorsed by the findings of the previous studies (LC50 > 10 μg/mL) [1,3].

Ref:

  1. Omeke, J. N.; Anaga, A. O.; Okoye, J. A. Brine shrimp lethality and acute toxicity tests of different hydro-methanol extracts of Anacardium occidentale using in vitro and in vivo models: a preliminary study. Clin. Path. 2018, 27, 1717-1721.
  2. Meyer, B. N.; Ferrigni, N. R.; Putnam, J. E.; Jacobsen, L. B.; Nichols, D. E.; Mc Laughlin, J. L. Brine shrimp: a convenient general bioassay for active plant constituents. Planta Medica. 1982, 45, 31–4.
  3. Rahman, M. A.; Sultana, R.; Emran, T. B.; Islam, M. S.; Rahman, M. A.; Chakma, J. S.; Rashid, H. U.; Hasan, C. M. Effects of organic extracts of six Bangladeshi plants on in vitro thrombolysis and cytotoxicity. BMC Complement Altern Med. 2013,13, 25.

  1. MEB and MEF have different IC50. Why did the authors use 200 mg/kg and 400 mg/kg for the animal test? How those dosages were established and based on? Suggest the authors must explain them well.

Authors’ responses: Thank you very much for your nice observations and comments. We have used two different doses (200 mg/kg body weight and 400 mg/kg body weight) in the animal study. We have chosen these doses based on current laboratory protocol and traditional guidelines. We have found zero mortality rate of the mice after administrating of the both doses of the crude. Besides, numerous studies have reported these two doses for preliminary pharmacological interventions of crude extracts derived from medicinal plants.    

  1. All the legends of the Tables did not describe well. Suggest must supplement experimental protocol, animal number, and statistical analysis.

Authors’ responses: Thank you very much for your suggestions. We have revised all the legends of the tables as follows: Also, we have corrected number of animals in each group. Kindly note that we have explained in detailed all the protocols and methods adopted in the study.

Table 1: Total phenol content, antioxidant potentials, and cytotoxic effects with IC50 (obtained from DPPH assay) and LC50 (obtained from brine shrimp lethality assay) values of MEB, MEF and different solvent fractions of W. coccinea.

Table 2: Anti-diarrheal effects in terms of % reduction of diarrheal feces following administrating the methanolic extract of bark (MEB) and methanolic extract of fruit (MEF) of W. coccinea in castor oil-induced diarrheal mice model.

Table 3: Hypoglycemic effect in terms of % reduction in blood glucose level of methanolic extract of bark (MEB) and methanolic extract of fruit coat (MEF) of W. coccinea on glucose-induced hyperglycemic mice model.

Table 4: Central analgesic effect of methanolic extract of bark (MEB) and methanolic extract of fruit coat (MEF) of W. coccinea in mice by tail immersion method.

Table 5: Peripheral analgesic effect of methanolic extract of bark (MEB) and methanolic extract of fruit coat (MEF) of W. coccinea in acetic acid-induced abdominal writhing mice.

Reviewer 3 Report

At first glance, I was interested in reading this manuscript. In this paper, authors reported the phytochemical and pharmacological investigations of the title plant. This theme is interesting, and basically they made a broad spectrum activity study for the extracts. However, this article had serious flaws and further experiments are needed.

In the first paragraph of Introduction, authors tried to give a brief introduction of medicinal plants and their pharmaceuticals. In this short part, eleven publications were cited, eight of which were authors’ (Md. Jamal Hossain and Mohammad A. Rashid) papers. Although they did much work on medicinal plants, there are many world famous scientists who made significant contributions in this field.

BTW, please update the Ref. [11] as ‘Newman, D. J.; Cragg, G. M., Natural products as sources of new drugs over the nearly four decades from 01/1981 to 09/2019. J. Nat. Prod., 2020, 83, 770-803.’.

On the second paragraph, authors presented the characteristics, distribution and name of the title plant, but didn’t give a introduction of previous phytochemical and pharmacological investigations of this species. For readers of this journal, they are more interested in the latter.

BTW, ‘Figure 1 presents different parts of the plant including the leaf, flower, fruit, and seed.’ This figure was missing.

What are the amounts of compounds 14? This manuscript is titled as ‘ Phytochemical Isolation of…’, however, in the Experiment part, only Sephadex LH-20 and PTLC were used to isolate the compounds from the extracts. As shown in this part, the resulting crude methanol extract of the bark, fruit coat, and seed pulp were quite a lot. Even if only the basic silica gel column chromatography used, dozens of compounds including new compounds could be obtained. In addition, the purified four compounds shown in Figure 1 are frequently encountered in different plants. Therefore, these compounds can’t represent the characteristic constituents of  Wrightia coccinea.

For the structural elucidation part, only the 1H NMR data were listed. How about the mass data? For identification, the purity of the tested compound is critical. As shown in their 1H NMR spectra (Figures S2, S4 and S5), the impurities were clearly visible. Further purification is required for these compounds. In addition, there are serious mistakes for the 1H NMR data. For example, the integration for the chemical shifts at δH ‘3.55, 3.30 (3H, m, J = 11 Hz, H3-28)’, ‘0.94 (3H, s, H3-29/H3-30)’, ‘0.86 (3H, s, H3-23/H3-26)’, the coupling patterns and constants at δH ‘3.55, 3.30 (3H, m, J = 11 Hz, H3-28)’, ‘7.93 (1H, d, J = 0.6, 7.6 Hz, H-5)’, ‘7.85 (1H, t, J = 1.2, 7.6 Hz, H-4′)’, ‘7.79 (1H, t, J = 1.2, 8.4 Hz, H-7)’, ‘7.68 (1H, t, J = 7.6, 8.0 Hz, H-5′)’, ‘7.43 (1H, t, J = 1.2, 8.0 Hz, H-6)’.

Why didn’t screen the antioxidant, cytotoxic, anti-diarrheal, hypoglycemic, central and peripheral analgesic properties for the four compounds?

Author Response

Reviewer 3

Comments and Suggestions for Authors

At first glance, I was interested in reading this manuscript. In this paper, authors reported the phytochemical and pharmacological investigations of the title plant. This theme is interesting, and basically they made a broad spectrum activity study for the extracts. However, this article had serious flaws and further experiments are needed.

Authors’ responses: I would like to thank you very much to reviewer 3 for reviewing the manuscript and providing your insightful comments on the manuscript. All the comments were carefully studied. We have improved and corrected the raised points accordingly. Thank you again for your kind efforts and time for reviewing the manuscript.  

In the first paragraph of Introduction, authors tried to give a brief introduction of medicinal plants and their pharmaceuticals. In this short part, eleven publications were cited, eight of which were authors’ (Md. Jamal Hossain and Mohammad A. Rashid) papers. Although they did much work on medicinal plants, there are many world famous scientists who made significant contributions in this field.

BTW, please update the Ref. [11] as ‘Newman, D. J.; Cragg, G. M., Natural products as sources of new drugs over the nearly four decades from 01/1981 to 09/2019. J. Nat. Prod.202083, 770-803.’.

Authors’ responses: Thank you very much for your comment. We are sorry. We should have more careful during citations. We have removed some previous self-citations and input some other prominent sources. Also, we have updated the reference number 11 as follows according to your suggestions:

Newman, D. J., Cragg, G. M. Natural Products as Sources of New Drugs over the Nearly Four Decades from 01/1981 to 09/2019. J. Nat. Prod. 2020; 83(3): 770-803.

On the second paragraph, authors presented the characteristics, distribution and name of the title plant, but didn’t give a introduction of previous phytochemical and pharmacological investigations of this species. For readers of this journal, they are more interested in the latter.

BTW, ‘Figure 1 presents different parts of the plant including the leaf, flower, fruit, and seed.’ This figure was missing.

Authors’ responses: Thank you very much for your comment. We are sorry. We have corrected as it was Figure S1 as follows:

Figure S1 presents different parts of the plant including the leaf, fruit, and seed. (Line: 75)

What are the amounts of compounds 14?

Authors’ responses: Thank you very much for your questions. We have found the four purified compounds (compounds 1 to 4) with 2 mg, 4 mg, 1.5 mg and 4 mg, respectively.

This manuscript is titled as ‘ Phytochemical Isolation of…’, however, in the Experiment part, only Sephadex LH-20 and PTLC were used to isolate the compounds from the extracts. As shown in this part, the resulting crude methanol extract of the bark, fruit coat, and seed pulp were quite a lot. Even if only the basic silica gel column chromatography used, dozens of compounds including new compounds could be obtained. In addition, the purified four compounds shown in Figure 1 are frequently encountered in different plants. Therefore, these compounds can’t represent the characteristic constituents of Wrightia coccinea.

Authors’ responses: I would like to thank you very much for your insightful observations and comments. You are right that we have isolated four compounds from the plant that have been also reported from several plants. Kindly, note that, as of our searching experience, this is the very first phytochemical investigation of the plant and first pharmacological interventions of the crude and fractions derived from the plant. Besides, this study will provide base-line data for further detailed investigations of various biological activities of the plant. Furthermore, among the four isolated compounds, wrightiadione might be a representative compound from the plant which is also reported for first time in this study, revealing the novelty of the research. 

For the structural elucidation part, only the 1H NMR data were listed. How about the mass data? For identification, the purity of the tested compound is critical. As shown in their 1H NMR spectra (Figures S2, S4 and S5), the impurities were clearly visible. Further purification is required for these compounds. In addition, there are serious mistakes for the 1H NMR data. For example, the integration for the chemical shifts at δH ‘3.55, 3.30 (3H, m, J = 11 Hz, H3-28)’, ‘0.94 (3H, s, H3-29/H3-30)’, ‘0.86 (3H, s, H3-23/H3-26)’, the coupling patterns and constants at δH ‘3.55, 3.30 (3H, m, J = 11 Hz, H3-28)’, ‘7.93 (1H, d, J = 0.6, 7.6 HzH-5)’, ‘7.85 (1H, t, J = 1.2, 7.6 HzH-4′)’, ‘7.79 (1H, t, J = 1.2, 8.4 Hz, H-7)’, ‘7.68 (1H, t, = 7.6, 8.0 Hz, H-5′)’, ‘7.43 (1H, t, J = 1.2, 8.0 Hz, H-6)’.

Authors’ responses: I would like to thank you very much for your comments. You are right that we have conducted 1H NMR during elucidation of structure of the compounds and we did not report the mass data. Kindly note that we have confirmed the purities of these compounds by utilizing thin layer chromatographic technique (TLC). During performing the NMR, the spectrum of compounds 1 & 3 showed few impurities along with the characteristic peaks. During first version of the manuscript, we did not provide the clear version of the spectrum. In the revised version of the manuscript, we are providing the clear version of the manuscript. We are acknowledging your observations regarding the few minor impurities. However, the published data and the characteristics peaks supported the reported compounds of this study. In addition, we have revised the 1H NMR data of compound 1 and 2 as per your concerns as follows:

3β-Acetyloxy-olean-12-en-28-ol (Compound 1): White solid crystal; 1H NMR (400 MHz, CDCl3): δ 5.30 (1H, br. s, H-12), 4.50 (1H, dd, J = 7.0, 10 Hz, H-3), 3.55 (2H, s, H3-28), 2.05 (3H, s, -OAc), 1.16 (3H, s, H3-27), 0.94 (6H, s, H3-29 and H3-30), 0.90 (3H, s, H3-25), 0.88 (3H, s, H3-24), 0.86 (3H, s, H3-23/H3-24) (Figure S2). (Lines: 351-154)

Wrightiadione (Compound 2): White amorphous powder; 1H NMR (400 MHz, CDCl3): δ 8.64 (1H, d, J = 8.0 Hz, H-8), 8.46 (1H, d, J = 8.0 Hz, H-5), 8.04 (1H, d, J = 8.0 Hz, H-3’), 7.93 (1H, d, J = 7.6 Hz, H-6’), 7.85 (1H, t, J = 7.2, 7.6 Hz, H-5’), 7.79 (1H, t, J = 7.2, 8.4 Hz, H-7), 7.68 (1H, t, J = 7.6, 8.0 Hz, H-4’), 7.43 (1H, t, J = 7.2, 8.0 Hz, H-6) (Figure S3).

The 1H NMR spectrum (400 MHz, CDCl3) of compound 2 gave resonances of four aromatic proton doublet at δ 8.64 (J = 8.0 Hz), 7.93 (J = 7.2 Hz), 8.04 (J = 8.0 Hz) and 8.46 (J = 8.0 Hz); four aromatic proton triplets at δ 7.79 (J = 1.2, 8.4 Hz), 7.43 (J = 7.2, 8.0 Hz), 7.85 (J = 7.2, 7.6 Hz) and 7.68 (J = 7.6, 8.0 Hz). The above signals revealed the presence of two aromatic rings with four adjacent protons in each ring, which indicated two isolated spin systems: H-5/H-6/H-7/H-8 and H-3′/H-4′/ H-5′/ H-6′. 1H NMR spectrum of compound 2 confirmed the aromatic rings without the hydroxy group. Compound 2 was characterized as wrightiadione [46]. (Lines: 363-374)

Why didn’t screen the antioxidant, cytotoxic, anti-diarrheal, hypoglycemic, central and peripheral analgesic properties for the four compounds?

Authors’ responses: I would like to thank you very much for your querries. It is a very relevant question. As we have found very low amounts of isolated compounds due to lower % yield, we could not screen these biological activities of the four compounds. However, we have conducted extensive in silico studies of these compounds against glutathione reductase (PDB ID: 3GRS), epidermal growth factor receptor (PDB ID: 1XKK), kappa opioid receptor (PDB ID: 6VI4), glucose transporter 3 (GLUT 3) (PDB ID: 4ZWB), Mu-opioid receptor (PDB ID: 5C1M), and cyclooxygenase 2 (COX-2) (PDB ID: 1CX2) proteins for assessing the antioxidant, cytotoxicity, antidiarrheal, hypoglycemic, central and peripheral analgesic activities, respectively.  

Round 2

Reviewer 1 Report

The study has been improved sufficiently. I approve this paper for publication

Author Response

Authors' responses: I would like to thank you very much to reviewer 1 for your kind effort and time in reviewing our manuscript. We are greatly grateful to you for your approval of our manuscript for publication.

Reviewer 2 Report

Review reports molecules-1680918-R1:

This manuscript's title is "Isolation of Phytoconstituents of Wrightia coccinea (roxb. Ex hornem.) Sims and Evaluation of its Antioxidant, Cytotoxicity, Antidiarrheal, Hypoglycemic, and Analgesic Properties Via In Vitro, In Vivo and In Silico Approaches" The authors intended to investigate The isolated compounds of Wrightia coccinea exhibited 1) In vitro antioxidant activity and cytotoxicity assay. 2.) In Vivo Anti-diarrheal, Hypoglycemic, and central and peripheral analgesic activity assay. 3.) In silico molecular docking binding affinity assay towards the active binding sites of glutathione reductase, epidermal growth factor receptor (EGFR), kappa opioid receptor (k-receptor), glucose transporter 3 (GLUT 3), Mu opioid receptor (m-receptor), and cyclooxygenase 2 (COX-2) proteins, respectively. However, this manuscript still has not well related and merged with the three-part into one well-prepared manuscript. Suggest some notes below that the authors must explain well.

Comments from the reviewer:

1.      Suggest a conclusion section at the end of the Abstract.

Authors’ responses:

Thank you very much for your kind suggestion. We have added the following lines in the abstract part.

Comment from the Reviewer:

The title spelling still had some errors and was too long. Suggest the authors check well.

2.      There were no results about seed pulp extract (MES). Suggesting cancel all the described sections related to MES fraction.

Authors’ responses:

Thank you very much for your comments. We have isolated three compounds (1 to 3) from methanol extract of bark and another compound was isolated from methanol extract of seed pulp (compound 4).

Comment from the Reviewer:

According to the authors described in the response manuscript, that there are three parts of methanolic extract of the plant including the (bark, fruit coat (MEF), and seed pulp).β-acetyloxy-olean-12-en-28-ol (1), wrightiadione (2), 22β-hydroxylupeol (3) were isolated from the bark methanol extract (MEB) and β-sitosterol was isolated from the seed pulp methanol extract (MES). MEF seems did not involve any active fraction. Why the authors did not use the MES but used the MEF in the anti-diarrheal, hypoglycemic, and central and peripheral analgesic activity assay in vivo. It was unreasonable and not well designed. Even contienue to process the molecular docking assay. Suggest the authors must explain them well. 

3.      (1), (2), (3), and β-sitosterol (4) were isolated from Wrightia coccinea. Suggesting the authors supplement the yield rate of each compound, isolated protocol or data (PTLC, VLC, or HPLC), and explain why did not use them or β-sitosterol (easy to obtain) as a positive control in vitro or in vivo study? 

Authors’ responses:

Thank you very much for your nice observations and comments. We have stated as follows and added two tables in the supplementary file (Table S1 and Table S2).

Comment from the Reviewer:

However, the authors described in the response manuscript, that there are three parts of methanolic extract of the plant including the (bark (MEB), fruit coat (MEF), and seed pulp (MES)). (1), (2), and (3) were isolated from the bark methanol extract and β-sitosterol was isolated from the seed pulp methanol extract. However, The author did not cite any reference but only supplemented the NMR spectra of compound identification information but still did not supplement the PTLC, VLC, or HPLC isolation information from Wrightia coccinea. It can not prove that three active compounds were isolated from Wrightia coccinea. Suggest the authors supplement the isolated or cited information and explain them well.

4.      In table 2, the authors have separated the methanolic extract (ME) into Petroleum ether (PE), dichloromethane (DCM), chloroform (CF), and aqueous (AQ) fraction. Which fraction contains those four compounds? Why did not use the bioactive fraction but used the crude methanolic extract study in vivo?

Authors’ responses:

Thank you very much for your comments. We have clearly mentioned in the method section that we have isolated three compounds 1 to 3 from the methanolic bark extract and compound 4 from the methanolic seed pulp extract of the plant. Therefore, in the animal study (in vivo investigation), we have used the crude extract.

Comment from the Reviewer:

There are three parts of methanolic extract of the plant including the (bark (MEB), fruit coat (MEF), and seed pulp (MES)). (1), (2), and (3) were isolated from the bark methanol extract and β-sitosterol was isolated from the seed pulp methanol extract. Those four fractions of Petroleum ether (PE), dichloromethane (DCM), chloroform (CF), and aqueous (AQ) fraction did not use in any pharmacological experiments. It seems extra data and did not connect anything to this manuscript. Suggest the authors cancel Table 1.

5.      Those four compounds were not phenolic compounds and could be not the chemical antioxidative compounds, why did the authors measure the total phenolic content assay and DPPH assay? It is the wrong direction. Otherwise, those pharmacological active compounds were from phenolic compounds, and those activities were not from those four compounds? Suggest the authors must explain well.

Authors’ responses:

Thank you very much for your comments. We have agreed with your comments that all these four compounds are not chemically phenolic compounds. We have investigated the total phenolic content (TPC) from the fractions derived from the fruit and bark methanolic extracts, and we have found promising results in the TPC and DPPH assay. Though we have isolated four compounds, and they are not phenolic. However, the aqueous fraction of the bark and chloroform fraction of the fruits provided the maximum amount of phenol content (220.62 and 190.83 mg of GAE/g of the dry extract, respectively). They also revealed the most potent antioxidant capacity (IC50 = 7.22 and 4.5 µg/mL, respectively) in DPPH free radical scavenging assay compared with the standard ascorbic acid (IC50 = 17.45 µg/mL). These results indicated that the fractious contain more antioxidant compounds and phenolic compounds that may have created a new indication for future prospective research. Besides, the in-silico studies revealed that the four compounds have more binding affinity towards glutathione reductase enzyme compared to the standard antioxidant drug BHT.

Comment from the Reviewer:

However, those four active compounds are not phenolic compounds, To analyze the total phenol content seems did not any meaning. Unless the authors found some bioactive phenolic compound in the extract in the Wrightia coccinea in this manuscript. t seems extra data and did connect anything for this manuscript. Suggest the authors cancel Table 1.

6.      In table 1, It is unbelievable, that IC50 was higher than LC50. What is the possibility that the authors can get those results? Suggesting the authors must explain them well.

Authors’ responses:

Thank you very much for your observations and comments. The IC50 values of the fractions were measured from the DPPH assay and LC50 values were measured from the brine shrimp lethality bioassay. For your kind perusal, we have stated here the modified legend of table 1 as follows:

Comment from the Reviewer:

In vivo study, used the different models to analyze IC50 and LC50 is innocence. However, it was necessary to use the mice to analyze the LC50 in the following in vivo study. Moreover, it was easy to calculate the IC50 from the following anti-diarrheal, hypoglycemic, and central and peripheral analgesic activity assays to compare the positive drug in vivo. It was not necessary to do any extra IC50 and LC50. t seems to have extra data and did not connect anything for this manuscript. Suggest the authors cancel Table 1.

7.      According to the pharmacotherapies concept, the therapeutic index LC50/IC50 is better lower than 10. The LC50 of MEB and MEF were very lower. It’s meaning that MEB and MEF seem to have a very high toxicity. Do they worthy to introduce to people? Suggesting the authors must explain them well.

Authors’ responses:

Thank you very much for your nice observations and comments. We have addressed and discussed the issue in the discussion part of the manuscript as follows:

As toxicity is a major concern for a crude drug, brine shrimp (Artemia salina) lethality bioassay, a cost-effective and reliable technique for preliminary screening of cytotoxicity, was conducted to predict the toxicity of the plant extracts [1]. Meyer et al. [2] reported that the cutoff point for bioactive phytochemicals is LC50 value less than 1000 μg/mL. The current study revealed that the measured LC50 values in the brine shrimp lethality bioassay were below 1000 μg/mL. No crude extract and fraction derived from the plant might be considered severely toxic or lethal, endorsed by the findings of the previous studies (LC50 > 10 μg/mL) [1,3].

Comment from the Reviewer:

In vivo study, using the different models to analyze IC50 and LC50 is unreasonable. However, it was necessary to use the mice to analyze the LC50 in the following study in vivo. Moreover, it was easy to calculate the IC50 from the following anti-diarrheal, hypoglycemic, and central and peripheral analgesic activity assays to compare the positive drug. It was not necessary to do any extra IC50 and LC50 assays. It seems the extra data did not relate anything to this manuscript. Suggest the authors can cancel Table 1.

8.      MEB and MEF have different IC50. Why did the authors use 200 mg/kg and 400 mg/kg for the animal test? How those dosages were established and based on? Suggest the authors must explain them well. 

Authors’ responses:

Thank you very much for your nice observations and comments. We have used two different doses (200 mg/kg body weight and 400 mg/kg body weight) in the animal study. We have chosen these doses based on current laboratory protocol and traditional guidelines. We have found zero mortality rate of the mice after administrating of the both doses of the crude. Besides, numerous studies have reported these two doses for preliminary pharmacological interventions of crude extracts derived from medicinal plants.

Comment from the Reviewer:

The authors seem not to have any principle for the dosage for in vivo tests. Still did not answer the comment well.

Moreover, in the antioxidant potentiality assay, BHT is the best in vitro assay but did not compare with any of those four active compounds. In silico binding affinity of BHT is -7.3 at the medium levels. It seems not a good standard of positive control. In the cytotoxicity assay, using vincristine is unreasonable. It is an anticancer drug it is not suitable to be a cytotoxicity agent. However, vincristine is the most cytotoxicity in vivo assay but did not compare with any of those four active compounds in vivo. In silico binding affinity of vincristine is -6.3 at the lowest cytotoxicity. It seems not a good standard of positive control. In the Anti-diarrheal assay, Loperamide is the best in vivo but did not compare with any of those four active compounds in vivo. In silico binding affinity of Loperamide is -7.3 at medium levels. It seems not a good standard of positive control. In the central analgesic activity assay, morphine is the best in vivo but did not compare with any of those four active compounds in vivo. In silico assay, the binding affinity of morphine is -8.0 at medium levels. β-Sitosterol is -9.7 is most binding affinity to the u-receptor? It is unbelievable. Suggest the authors must check them well. In the peripheral analgesic activity assay, diclofenac is the best in vivo but did not compare with any of those four active compounds in vivo. In the silico binding affinity of diclofenac is -7.0 at medium levels. It seems not a good standard of positive control. β-Sitosterol is -9.7 is most binding affinity to the COX-2 enzyme. It is unbelievable for the readers. Suggest the authors must check those related binding affinities in the docking models well. Moreover, suggest the authors better supplement those active compound data for the pharmacological in vivo study to confirm and compare the docking models.

9.      All the legends of the Tables did not describe well. Suggest must supplement experimental protocol, animal number, and statistical analysis.  

Authors’ responses:

Thank you very much for your suggestions. We have revised all the legends of the tables as follows: Also, we have corrected number of animals in each group. Kindly note that we have explained in detailed all the protocols and methods adopted in the study.

Comment from the Reviewer:

No more comments for this viewpoint.

Author Response

Kindly check the uploaded PDF (responses to reviewer 2)

Reviewer 3 Report

Only some comments has been addressed and the paper is revised accordingly, additional and ongoing revisions are required.

1. For the purity, TLC can’t confirm the purity firmly. For the compounds whose structures and polarities are closely similar, they will be observed as one spot. As shown in the 1H NMR spectra, the peaks for the purities could be easily observed and their integrations were not small. Without any other spectra, only select some peaks for structural elucidation is not reliable.

2. For structural elucidation part, more efforts are required. Firstly, add the mass data for each compounds, which could be used to check the molecular formula, if there is no 13C NMR data. Secondly, check the NMR data more carefully. For compound 1, the integration for the chemical shifts δH 0.88 of two methyls is wrong. And two different chemical shifts were wrongly assigned for H3-24. For compound 2, the mistake can be easily found such as two constants were attributed for one triplet at δH 7.85. Thirdly, the expression for NMR analysis should be modified including the grammar errors, such as ‘four aromatic proton doublet’. Except for these two compounds, the NMR data for compound 3 should be checked especially for the peak patterns at δH 4.56. The NMR data for compound 4 should be added, which is crucial for structure elucidation.

3. Modify the references’ styles according to the Journal’s Guidelines.

Author Response

Reviewer 3

Comments and Suggestions for Authors

Only some comments has been addressed and the paper is revised accordingly, additional and ongoing revisions are required.

  1. For the purity, TLC can’t confirm the purity firmly. For the compounds whose structures and polarities are closely similar, they will be observed as one spot. As shown in the 1H NMR spectra, the peaks for the purities could be easily observed and their integrations were not small. Without any other spectra, only select some peaks for structural elucidation is not reliable.

Author responses: Thank you very much for the comments. We have acknowledged that the compounds may contain few impurities that is visible in NMR spectrum. Though we have observed some impurities during NMR spectrometry, we have found major characteristics peaks, which were comparable with previously published data for structure elucidation of our targeted compounds. As we have confirmed these compounds with the previous reported data, we have not conducted any other spectra.  

  1. For structural elucidation part, more efforts are required. Firstly, add the mass data for each compounds, which could be used to check the molecular formula, if there is no 13C NMR data. Secondly, check the NMR data more carefully.

Author responses: Thank you very much for the comments. We have not conducted the mass analysis of the compounds. These are the reported compounds which can be easily compared with the published NMR data. Besides, we have revised and corrected the NMR data according to your valuable suggestions.   

For compound 1, the integration for the chemical shifts δH 0.88 of two methyls is wrong. And two different chemical shifts were wrongly assigned for H3-24.

Author responses: Thank you very much for the comments. The integration value δH 0.88 should be for one methyl (-CH3) group, but due to the minor impurities, it might have been distorted. Besides, we have corrected the chemical shift assigned for H3-26 (Lines: 344). Sorry for the typos error.    

For compound 2, the mistake can be easily found such as two constants were attributed for one triplet at δH 7.85. Thirdly, the expression for NMR analysis should be modified including the grammar errors, such as ‘four aromatic proton doublet’.

Author responses: Thank you very much for the comments. In case of compound 2, we have carefully checked and removed the double constants from the NMR data reporting. Moreover, we have corrected the grammar errors, such as ‘four aromatic proton doublets’ (Lines: 358)

Except for these two compounds, the NMR data for compound 3 should be checked especially for the peak patterns at δH 4.56.

Author responses: Thank you very much for the comments. For compound 3, δH 4.56 should show the integration value for 1H, however it is slightly low, which is an exception and it might be machine limitation. 

The NMR data for compound 4 should be added, which is crucial for structure elucidation.

Thank you very much for your comments. We have stated the NMR data of the compound 4 as follows:

β-Sitosterol:  White amorphous powder; 1H NMR (400 MHz, CDCl3): d 0.682 (3H,s, Me -18), 0.81 (3H, d, J = 7.0 Hz, Me-27), 0.83 (3H, d, J = 7.0 Hz, Me-­26), 0.85 (3H,t, J = 7.0 Hz, Me -29), 0.92 (3H, d, J = 6.4 Hz, Me -21), 1.01 (3H, s, Me -19), 3.53 (1H, p, H - 3a ), 5.35 (1H,d br, J = 3.6 Hz, H­6). (Lines: 374-377)

  1. Modify the references’ styles according to the Journal’s Guidelines.

Author responses: Thank you very much for the comments. We have corrected the references as per the journal guidelines.

Round 3

Reviewer 2 Report

Review reports molecules-1680918-R33:

This manuscript's title is "Chemical and Pharmacological Profiling of Wrightia coccinea (roxb. Ex hornem.) Sims Focusing Antioxidant, Cytotoxicity, Antidiarrheal, Hypoglycemic, and Analgesic Properties" The authors intended to conduct the phytochemical and pharmacological investigations of the bioactive compounds from Wrightia coccinea through in vitro, in vivo, and silico models. However, this manuscript still can be improved.

Comments from the reviewer:

1. The author still did not supplement enough chromatographic isolation information to prove that β-acetyloxy-olean-12-en-28-ol, wrightiadione, and 22β-hydroxylupeol were isolated from Wrightia coccinea. Suggesting the authors supplement enough chromatographic evidence of compounds in isolation such as TLC, PTLC, or HPLC spectra to prove that those compounds were isolated from Wrightia coccinea.

2. No more extra comments.

Author Response

Reviewer 2

Comments and Suggestions for Authors

Review reports molecules-1680918-R3:

This manuscript's title is "Chemical and Pharmacological Profiling of Wrightia coccinea (roxb. Ex hornem.) Sims Focusing Antioxidant, Cytotoxicity, Antidiarrheal, Hypoglycemic, and Analgesic Properties" The authors intended to conduct the phytochemical and pharmacological investigations of the bioactive compounds from Wrightia coccinea through in vitro, in vivo, and silico models. However, this manuscript still can be improved.

Authors responses: I would like to thank you very much to reviewer 2 for your further comments to improve the manuscript. We are grateful to you for your kind efforts and time for reviewing our manuscript and providing your valuable feedback.   

Comments from the reviewer:

  1. The author still did not supplement enough chromatographic isolation information to prove that β-acetyloxy-olean-12-en-28-ol, wrightiadione, and 22β-hydroxylupeol were isolated from Wrightia coccinea. Suggesting the authors supplement enough chromatographic evidence of compounds in isolation such as TLC, PTLC, or HPLC spectra to prove that those compounds were isolated from Wrightia coccinea.

Authors' responses: Thank you very much for your suggestions. We have previously added the PTLC data. In this revised version, we are also proving the TLC data in Figure S2 as follows in the supplementary file for proving that these compounds were isolated from W. coccinea plant.

Figure S2: (A) Initial Screening of the VLC fractions of Wrightia coccinea (10% ethyl acetate in hexane), (B) screening of the mixed similar beakers VLC fractions of Wrightia coccinea (15% ethyl acetate in hexane).

  1. No more extra comments.

Authors' responses: Thank you very much for your kind feedback.

Reviewer 3 Report

The authors have revised the manuscript according to the comments. 

Author Response

Authors' responses: I would like to thank you very much to reviewer 3 for your kind effort and time in reviewing our manuscript. We are greatly grateful to you for your kind approval.
